# LETS Forecast: Learning Embedology for Time Series Forecasting

**Abrar Majeedi** [1]   **Viswanatha Reddy Gajjala** [1 2]   **Satya Sai Srinath Namburi GNVV** [2]   **Nada Magdi Elkordi** [2]
**Yin Li** [1 2]

## Abstract

Real-world time series are often governed by complex nonlinear dynamics. Understanding these underlying dynamics is crucial for precise future prediction. While deep learning has achieved major success in time series forecasting, many existing approaches do not explicitly model the dynamics. To bridge this gap, we introduce DeepEDM, a framework that integrates nonlinear dynamical systems modeling with deep neural networks. Inspired by empirical dynamic modeling (EDM) and rooted in Takens' theorem, DeepEDM presents a novel deep model that learns a latent space from time-delayed embeddings, and employs kernel regression to approximate the underlying dynamics, while leveraging efficient implementation of softmax attention and allowing for accurate prediction of future time steps. To evaluate our method, we conduct comprehensive experiments on synthetic data of nonlinear dynamical systems as well as real-world time series across domains. Our results show that DeepEDM is robust to input noise, and outperforms state-of-the-art methods in forecasting accuracy. Our code is available at: https://abrarmajeedi.github.io/deep_edm.

## 1. Introduction

Time series forecasting is fundamental across multiple domains including economics, energy, transportation, and meteorology, where accurate predictions of future events guide critical decision-making. Deep learning has recently emerged as the dominant approach, driven by its ability to leverage large datasets and capture intricate nonlinearity. While deep models excel in prediction accuracy, they often

treat time series data as abstract patterns, and fall short in considering the underlying processes that generate them. Addressing this blind spot of deep models is important, because at its core, time series data is not merely sequences of numbers; rather, these data represent the dynamic behavior of complex systems, encoding the interplay of various factors over time. Indeed, many real-world time series data can be treated as manifestations of time-variant dynamics (Brunton et al., 2022). Therefore, understanding the underlying systems can unlock more effective forecasting strategies.

Dynamical systems modeling characterizes the evolution of deterministic or stochastic processes governed by underlying dynamics, thereby offering an appealing solution for time series forecasting. However, if a system is not specified, forecasting requires solving the challenging problem of inferring the underlying dynamics from observations. To address this, Empirical Dynamical Modeling (EDM) (Sugihara & May, 1990), a data-driven approach built on Takens' theorem (Takens, 1981b; Sauer et al., 1991), was developed to recover nonlinear system dynamics from partial observations of states. EDM leverages time-delayed embeddings to topologically reconstruct the system's state space from observed time series, which can then be used for forecasting. While EDM has demonstrated success in real-world applications (Ye et al., 2015; Sugihara et al., 2012), it assumes noise-free data, requires separate modeling for individual sequences, and imposes constraints over its forecasting horizon, significantly limiting its broader practical applicability.

To bridge the gap, we propose a novel framework—DeepEDM that integrates EDM and deep learning, addressing EDM's key limitations and introducing a new family of deep models for time series forecasting. Specifically, DeepEDM constructs time-delayed version of the input sequence, and projects them into a learned latent space that is more robust to noise. It further employs kernel regression implemented using highly efficient softmax attention (Vaswani et al., 2017), followed by a learned decoder, to model the latent dynamics and predict future values. Importantly, DeepEDM is fully differentiable, and thus can be learned end-to-end from large-scale data.

DeepEDM connects traditional EDM and modern deep learning. On one hand, it significantly extends EDM by im-

---

[1]Department of Biostatistics and Medical Informatics, University of Wisconsin-Madison [2]Department of Computer Sciences, University of Wisconsin-Madison. Correspondence to: Yin Li <yin.li@wisc.edu>.

*Proceedings of the 42$^{nd}$ International Conference on Machine Learning*, Vancouver, Canada. PMLR 267, 2025. Copyright 2025 by the author(s).

proving robustness against measurement noise, enabling the learning of a single parametric model to generalize across sequences, and supporting longer forecasting horizons. On the other hand, it integrates the rigor of dynamical systems modeling with the flexibility and scalability of deep learning, leading to a variant of Transformer model for time series forecasting, and providing theoretic insights for other Transformer-based models (Liu et al., 2024a; Nie et al., 2023; Chen et al., 2025).

Our **main contributions** are thus three folds. *First*, we propose DeepEDM, a novel framework inspired by dynamical systems modeling that leverages time-delayed embeddings for time series forecasting. *Second*, DeepEDM, grounded in Takens' theorem, addresses key limitations of EDM, and sheds light on prior Transformer-based time series models. *Third*, extensive experiments on synthetic datasets and real-world benchmarks, demonstrate state-of-the-art forecasting performance of DeepEDM.

## 2. Related Work

### 2.1. Deep Learning for Times Series Forecasting

There has been major progress in time series forecasting thanks to deep learning. Early approaches predominantly consider Recurrent Neural Networks (RNNs), especially Long Short-Term Memory (LSTM) networks (Hochreiter & Schmidhuber, 1997; Yu et al., 2017), which are adept at capturing long-term dependencies. Subsequent developments, such as LSTNet (Lai et al., 2018) and DeepAR (Salinas et al., 2020), integrate recurrent and convolutional structures to enhance forecasting accuracy. Temporal Convolutional Networks (TCNs) (Bai et al., 2018), and methods like MICN (Wang et al., 2023a) and TimesNet (Wu et al., 2023), leverage multi-scale information and adaptive receptive fields, improving multi-horizon forecasting capabilities. Recent works find that Multi-Layer Perceptrons (MLPs) can achieve competitive performance. Notably, TimeMixer (Wang et al., 2024a) presents a sophisticated MLP-based architecture that incorporates multi-scale mixing, outperforming previous MLP models such as DLinear (Zeng et al., 2023) and RLinear (Li et al., 2023).

Transformer-based (Vaswani et al., 2017) models have shown to be highly effective for long-term forecasting (Chen et al., 2025). Architectures like Reformer (Kitaev et al., 2020), Pyraformer (Liu et al., 2021), Autoformer (Wu et al., 2021), and Informer (Zhou et al., 2021) have enhanced the scalability and efficiency of attention mechanisms, adapting them for longer range time series forecasting. Subsequent innovations such as PatchTST (Nie et al., 2023), which proposes a patching-based channel independent approach along with instance normalization (Ulyanov et al., 2016), and iTransformer (Liu et al., 2024a), which utilizes a

channel-wise attention framework, have further improved the forecasting performance of attention-based models.

### 2.2. Learning Dynamical Systems for Forecasting

Learning dynamical systems for time series forecasting has garnered considerable interest within the research community. Many prior works builds on Koopman's theory (Mezić, 2021; Brunton et al., 2022), which represents a nonlinear system with a linear operator in an infinite-dimensional space. Examples includes Koopman Autoencoder (Lusch et al., 2018; Takeishi et al., 2017) and K-Forecast (Lange et al., 2021). Both approximate the Koopman operator in a high-dimensional space to effectively model nonlinear dynamics. These approaches enable scalable forecasting for complex systems by simultaneously learning the measurement function and the Koopman operator.

Recent developments, including Koopa (Liu et al., 2024b) and Deep Dynamic Mode Decomposition (Deep-DMD) (Alford-Lago et al., 2022), extend this framework. Koopa enhances the forecasting of nonlinear systems through a modular Fourier filter combined with a Koopman predictor. Together, these components hierarchically disentangle and propagate time-invariant and time-variant dynamics. DeepDMD employs deep learning to traditional DMD, facilitating the identification of coordinate transformations that linearize nonlinear system dynamics, thus capturing complex, multiscale dynamics effectively. A more recent work, Attraos (Hu et al., 2024), has explored alternative perspectives through chaos theory and attractor dynamics.

### 2.3. Empirical Dynamical Modeling

EDM (Chang et al., 2017; Sugihara & May, 1990) presents an alternative approach to model nonlinear dynamics. Rooted in Takens' theorem (Takens, 1981b), it relies on delay-coordinate embeddings to reconstruct the underlying attractor, thereby preserving the essential topological properties of the original dynamical system. Unlike Koopman, which linearizes nonlinear dynamics in a carefully chosen high dimensional space (approximation to an infinite-dimensional space), EDM can topologically reconstruct system dynamics using low dimensional observations, or even with a scalar observation at each time step (Takens, 1981b; Sauer et al., 1991). EDM is thus particularly attractive for problems with limited observation of the system states.

A line of prior research (Mezić & Banaszuk, 2004; Arbabi & Mezić, 2017; Mezić, 2022) has also explored the connection between Takens' theory and Koopman operator theory. Hankel-DMD (Arbabi & Mezić, 2017) applies dynamic mode decomposition to time-delayed time series, effectively performing Koopman spectral analysis on Takens' embeddings. Numerical issues associated with Hankel-DMD have been discussed in (Mezić, 2022).

## 2.4. Chaotic Time Series Forecasting

A related research direction focuses on forecasting chaotic time series via state space reconstruction, mirroring the underlying principles of EDM. Pioneering work by (Farmer & Sidorowich, 1987) introduced local approximation techniques within reconstructed state spaces using delay embeddings for short-term predictions. Subsequent studies explored feedforward neural networks for learning direct mappings from reconstructed phase states to future states (Karunasinghe & Liong, 2006). Recurrent models, especially Echo State Networks (ESNs) (Jaeger & Haas, 2004), and variants like robust ESNs (Li et al., 2012), further improved resilience to noise and outliers.

# 3. DeepEDM for Time Series Forecasting

We consider time series generated by discrete-time nonlinear dynamical systems, though all derivations can be readily extended to continuous-time systems. A discrete-time nonlinear dynamical system is defined as a recurrence relation in which a nonlinear function $\boldsymbol{\Phi}$ governs the evolution of the state variables $x_t \in \mathbb{R}^d$ at time step $t$:

$$x_{t+1} = \boldsymbol{\Phi}(x_t). \tag{1}$$

Oftentimes, the states of the system $x_t$ can not be directly observed and the governing equation $\boldsymbol{\Phi}$ is unknown. Instead, a common assumption is that measurements $y_t$ of the states $x_t$ can be acquired using

$$y_t = \boldsymbol{h}(x_t) + \epsilon, \tag{2}$$

where $\boldsymbol{h}$ is an *unknown* measurement function that maps a system state $x_t$ to its observation $y_t$ with a time-invariant stochastic noise $\epsilon$.

Our goal is time series forecasting, i.e., predicting future observations $y_{T+1:T+H}$ based on existing ones $y_{1:T}$. $y_{1:T}$ often referred to as *the lookback window* with length $T$, and $y_{T+1:T+H}$ as predictions with its *forecasting horizon $H$*. Without knowing the governing equation $\boldsymbol{\Phi}$ or the measurement function $\boldsymbol{h}$, this forecasting problem is very challenging even with a small amount of noise $\epsilon$. In what follows, we introduce the theoretic background, present our approach, and describe its practical instantiation.

## 3.1. Preliminaries: Takens' Theorem and EDM

### Takens' Theorem

Takens' theorem establishes the feasibility to "recover" the underlying dynamics defined by $\boldsymbol{\Phi}$, without knowing the observation function $\boldsymbol{h}$ and assuming zero noise (i.e., $\epsilon = 0$). In this case, forecasting becomes straightforward and only involves forwarding the uncovered dynamics. Intuitively, the theorem states that if $\boldsymbol{\Phi}$, $\boldsymbol{h}$ and the state space of $x$

are constrained, the dynamics can be topologically reconstructed, perhaps surprisingly, even with univariate measurements $y_{1:t}$. With slight abuse of notations, we now restate Takens' theorem in our setting.

**Theorem 3.1.** (Takens, 1981a) *Let $\mathcal{M}$ be a compact manifold of dimension $d$ defining the space of states $x$ and assume the observed time series data is univariate, i.e., $y \in \mathbb{R}$. For pairs of dynamics $\boldsymbol{\Phi}$ and observation function $\boldsymbol{h}$, where $\boldsymbol{\Phi} : \mathcal{M} \rightarrow \mathcal{M}$ is a $C^2$ smooth diffeomorphism, i.e., $\boldsymbol{\Phi}$ must be bijective and both $\boldsymbol{\Phi}$ and its inverse $\boldsymbol{\Phi}^{-1}$ are $C^2$ smooth, and $h : \mathcal{M} \rightarrow \mathbb{R}$ is a $C^2$ smooth function, it is a generic property that $\mathcal{H}(\boldsymbol{\Phi}, \boldsymbol{h}) : \mathcal{M} \rightarrow \mathbb{R}^{2d+1}$ defined by $\left(\boldsymbol{h}(x), \boldsymbol{h}(\boldsymbol{\Phi}(x)), \boldsymbol{h}(\boldsymbol{\Phi}^2(x)), ..., \boldsymbol{h}(\boldsymbol{\Phi}^{2d}(x))\right)$ is an immersion, i.e., $\mathcal{H}$ is injective and both $\mathcal{H}$ and $\mathcal{H}^{-1}$ are differentiable.*

The $(2d+1)$-D vectors $\left\{\left(\boldsymbol{h}(x), \boldsymbol{h}(\boldsymbol{\Phi}(x)), ..., \boldsymbol{h}(\boldsymbol{\Phi}^{2d}(x))\right)\right\}$ thus preserve the topology of the states $z_t$. By setting $x = x_{t-2d}$, it is easy to note that these vectors are $\left\{[y_{t-2d}, y_{t-2d+1}, \ldots, y_t]\right\}$, i.e., a time-delayed version of the observed time series. The theorem thus states that given a time series of 1D measurement $y_t$, its time delayed version $\hat{y}_{1:t}, \hat{y}_t = (y_{t-2d}, y_{t-2d+1}, ..., y_t)$ has a similar topology with the states $x_{0:t}$. Therefore, we can instead model the induced dynamics of $\hat{y}_{0:t}$ to recover properties of the underlying dynamics of $x_{1:t}$, as illustrated in Figure 1(a).

It is worth noting that Takens' theorem has two restrictive assumptions: (1) the state space must be a compact manifold; and (2) measurements are univariate. Recent developments have extended the theorem to more general settings, accounting for the state space as a compact invariant set within finite Euclidean space (Sauer et al., 1991), or the measurements as multivariate vectors (Deyle & Sugihara, 2011).

### Empirical Dynamic Modeling (EDM)

Built on Takens' theorem, EDM (Sugihara & May, 1990; Dixon et al., 1999; Sugihara et al., 2012; Chang et al., 2017) provides a computational method to reconstruct a system's state space from time series of its univariate measurements. We now briefly describe EDM with Simplex projection, which lays the foundation for our approach.

Simplex projection assumes univariate measurements $y_{1:T}$ and considers its time-delayed version $\hat{y}_{1:T}$ with $\hat{y}_t \in \mathbb{R}^{2d+1}$, i.e., time-delayed by $2d + 1$ steps. To forecast a future time step $y_{T+\Delta t}$ ($\Delta t \geq 1$), it first finds $2d + 2$ nearest neighbors $\{\hat{y}_{N_i}\}, i \in [1, ..., 2d + 2]$ for $\hat{y}_T$ using a pre-specific similarity metric, i.e., a kernel function $k(\hat{y}, \hat{y}')$[1]. These nearest neighbors $\{\hat{y}_{N_i}\}$ are assumed to define a *simplex* in the $2d+1$-D space of $\hat{y}$, i.e., a geometric structure that generalizes a triangle (in 2D) or a tetrahedron (in 3D)

---

[1]The original method in (Sugihara & May, 1990) used the radial basis function kernel $k(\hat{y}, \hat{y}') = \exp(-\|\hat{y} - \hat{y}'\|^2/2\sigma^2)$.

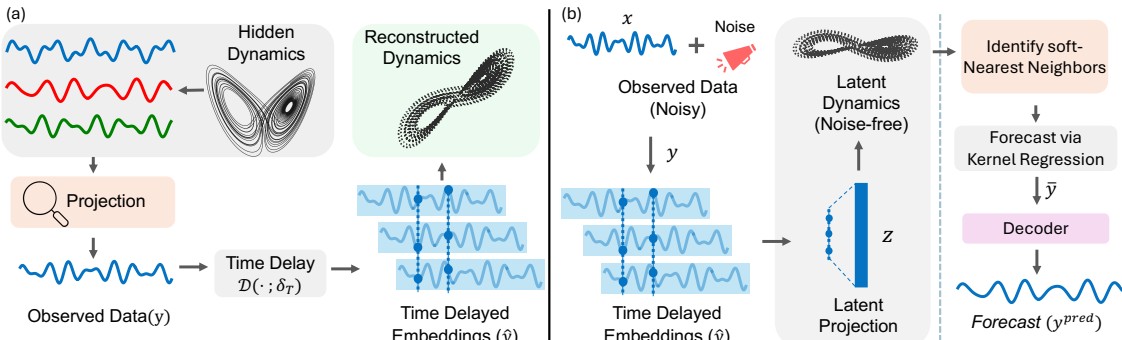

*Figure 1.* (a) **Takens' theorem in action**. The state space of an unknown nonlinear dynamical system is reconstructed using time-delayed embeddings from observed time series measurements (noise free). (b) **Overview of DeepEDM**. Time-delayed embeddings are constructed to model the system's underlying state space. These embeddings are then mapped into a learned latent space that is robust to measurement noise. Forecasting is performed via kernel regression followed by a learned decoder, where soft nearest neighbors for regression are defined in the latent space. This model, resembling the key idea of EDM, is fully differentially and thus can be learned from end-to-end.

to arbitrary dimensions. $y_{T+\Delta t}$ is then predicted by a linear re-weighting on this simplex, given by

$$y^{\text{pred}}_{T+\Delta t} = \frac{1}{\sum_{i=1}^{2d+2} w_i} \sum_{i=1}^{2d+2} w_i \cdot y_{N_i+\Delta t}, \qquad (3)$$

where the weight is given by $w_i = k\left(\hat{y}_T, \hat{y}_{N_i}\right)$. Again, $N_i$ indexes the $2d+2$ nearest neighbors of $\hat{y}_t$, and $y_{N_i+\Delta t}$ denotes the observed data $\Delta t$ steps after $N_i$. We note that Equation (3) can be viewed as the Nadaraya-Watson estimator using $2d+2$ nearest neighbors, where the regressor connects the input of $\hat{y}_{N_i}$ to its output of $y_{N_i+\Delta t}$.

Simplex projection can be also considered as a locally linear approximation to the manifold of the time-delayed observations $\hat{y}$, which is topologically equivalent to the state space of $x$. The key assumption is that $\hat{y}_{1:T}$ sufficiently covers the manifold, such that the nearest neighbors $\{\hat{y}_{N_i}\}$ of $\hat{y}_T$ correspond to underlying states similar to $x_t$. This assumption allows an empirical approximation of forwarding $\mathbf{\Phi}$ for forecasting, using the the future data of these nearest neighbors $(y_{N_i+\Delta t})$. However, it also imposes a practical constraint: the forecasting horizon $(H)$ must be significantly shorter than the length of the lookback window $(T)$.

### 3.2. Our Approach: DeepEDM

Despite its success (Ye et al., 2015; Sugihara et al., 2012), EDM with Simplex projection has three key limitations. First, it assumes noise-free measurements, leading to significant performance degradation when forecasting in the presence of noise. Second, it models each sequence independently, disregarding patterns shared across time series. Third, it imposes a constraint that the forecasting horizon must be much shorter than the lookback window.

To address these limitations, we present **DeepEDM**, a novel deep model that builds on the key idea of EDM, leveraging strengths from both paradigms. DeepEDM, as shown in Figure 1(b), consist of (1) a base forecasting model that gen-

erates initial predictions, relaxing the constraint on forecasting horizon; (2) a learned encoder to embed time-delayed time series into a latent space, gaining robustness against input noise; (3) a kernel regression to predict future data in the latent space, re-assembling Simplex projection while allowing for efficient and differentiable implementation; and (4) a decoder to output the final predictions and mitigate noise. Collectively, *DeepEDM is fully differentiable and enables end-to-end learning of a single parametric model for forecasting that generalizes across time series, avoiding per-sequence modeling in EDM*.

To simplify our notations, we describe DeepEDM in the context of univariate time series forecasting in this section. For multivariate time series, DeepEDM is applied *channel-wise*, meaning a single DeepEDM model is shared across individual variates — a strategy widely used in prior works (Nie et al., 2023; Zeng et al., 2023). We now introduce individual components, present the training scheme, and discuss links to Transformer-based models.

### Modeling

**Initial prediction**. DeepEDM starts with a simple, base prediction model $f(\cdot)$ (e.g., a linear model or an MLP). $f(\cdot)$ takes the input of the lookback window $y_{1:T}$ with $y_i \in \mathbb{R}$ (univariate or a single channel in a multivariate time series), and outputs the predictions $y^p$ for $H$ steps

$$y^p_{T+1:T+H} = f(y_{1:T}). \qquad (4)$$

This initial prediction allows us to concatenate the lookback window $y_{1:T}$ and the predicted window $y^p_{T+1:T+H}$, forming a new time series $[y_{1:T}, y^p_{T+1:T+H}]$. DeepEDM will now operate on this extended sequence and further refine the initial prediction, bypassing EDM's constraint on the forecasting horizon. This is particularly helpful for long-term forecasting, where $T$ might be smaller than $H$. **Time delay and encoding**. DeepEDM further time-delays the extended sequence $[y_{1:T}, y^p_{T+1:T+H}]$, and considers a learned

encoder Enc($\cdot$) to project the time-delayed signals into a latent space. Formally, this is given by

$$\hat{y}_{1:T+H} = \mathcal{D}([y_{1:T}, y^p_{T+1:T+H}]; \delta_T),$$
$$z_{1:T+H} = \text{Enc}(\hat{y}_{1:T+H}),$$
(5)

where $\mathcal{D}(\cdot; \delta_T)$ denotes a time delay operator with $\delta_T$ delay steps—a hyperparameter of DeepEDM. $\hat{y}_t$ is thus the time-delayed embedding of the concatenated sequence. Note that zero padding is added before the sequence to preserve the temporal dimension. The encoder Enc($\cdot$) is realized using a neural network with learnable parameters. Enc($\cdot$) is designed to extract features from the time-delayed embeddings of an input sequence, enabling meaningful comparisons among these embeddings with noisy measurements.

**Simplex projection with kernel regression**. DeepEDM further employs kernel regression for prediction, extending the key idea of Simplex projection in EDM. While Simplex projection finds $K(= \delta_t + 1)$ nearest neighbors — an operation that is not differentiable, we propose to instead leverage all data points, again using the Nadaraya–Watson estimator. In this case, we rely on the choice of the kernel $k(\cdot, \cdot)$ to down-weight irrelevant data. Formally, this is expressed as

$$\bar{y}_{t'+\Delta t} = \frac{1}{\sum_{t=1}^{T} k(z_t, z_{t'})} \sum_{t=1}^{T} k(z_t, z_{t'}) \cdot \hat{y}_{t+\Delta t},$$
(6)

where $t' \in [T, T + H - \Delta t]$ and we simply set $\Delta t = 1$ for a single-step forward prediction. We choose $k(z_t, z_{t'}) = \exp(\langle z_t, z'_t \rangle / \tau)$ with $\tau$ to control its decay, and leverage highly optimized softmax attention for efficient implementation (with $\tau$ as the temperature in softmax). While $\tau$ can be learned from data, we empirically find that doing so has minimal impact on overall performance, and keep $\tau = 1$.

Notably, unlike Simplex projection (Eq. 3), which predicts a scalar corresponding to a single step in a univariant time series, our kernel regression (Eq. 6) predicts a vector of size $\delta_T$ representing the time-delayed version of the time series.

**Prediction decoding**. Finally, DeepEDM decodes the output $y^{\text{pred}}_{T+1:T+H}$ based on $\bar{y}_{T+1:T+H}$ using a decoder Dec($\cdot$)

$$y^{\text{pred}}_{T+1:T+H} = \text{Dec}(\bar{y}_{T+1:T+H}),$$
(7)

where Dec($\cdot$) is realized with a lightweight neural network with learnable parameters. Dec($\cdot$) learns to reconstruct the predicted time series from its time-delayed version and, crucially, denoises the output to mitigate the effects of measurement noise introduced during kernel regression.

**Training**

DeepEDM includes learnable parameters in the base model $f(\cdot)$, the encoder Enc($\cdot$), and the decoder Dec($\cdot$). Our learning objective is to jointly optimize these parameters to minimize prediction errors on the training set. Omitting subscripts for simplicity, our training loss is defined as:

$$\mathcal{L} = \lambda \underbrace{\|y - y^{\text{pred}}\|_p}_{\mathcal{L}_{\text{err}}} + (1 - \lambda) \underbrace{\|\nabla y - \nabla y^{\text{pred}}\|_p}_{\mathcal{L}_{\text{td}}},$$
(8)

where $\nabla$ denotes the first order finite difference and $\lambda$ is the balancing coefficient. Namely, our loss minimizes the $L_p$ norm of the prediction errors ($\mathcal{L}_{\text{err}}$) and its temporal differences ($\mathcal{L}_{\text{td}}$). We also find it helpful to consider an adaptive $\lambda$ following (Xiong et al., 2024), especially for long-term forecasting problems. Further details of our loss function can be found in the Appendix A.2.

**Discussion**

**Relationships to Transformer-based models**. DeepEDM shares a strong conceptual connection with Transformer models widely used in time series forecasting. Specifically, the notion of time-delay embedding in EDM and DeepEDM can be viewed as a special case of local window patching (Nie et al., 2023). Moreover, the combination of encoder, kernel regression, and decoder resembles the structure of a Transformer block with self-attention (Chen et al., 2025), albeit with distinct definitions of queries, keys, and values. From this perspective, DeepEDM can be interpreted as a Transformer-like model with input patching, which refines initial predictions from a simple base model. Indeed, patching, Transformer architectures, and cascaded prediction have all been proven to be highly effective for time series forecasting.

### 3.3. Model Instantiation

**Base prediction model, encoder, and decoder**. The *base predictor* $f(\cdot)$ is realized using a multilayer perceptron (MLP) shared across all variates. Given historical input $y \in \mathbb{R}^{D \times T}$, $f(\cdot)$ maps $d$-th variate's time series $y_d \in \mathbb{R}^T$, to a prediction $y^p_d \in \mathbb{R}^H$, yielding the initial prediction $y^p \in \mathbb{R}^{D \times H}$. Subsequently, the lookback $y$ and the initial forecast $y^p$ are concatenated and time delayed by $\delta_T$ steps, resulting in $\hat{y}_{1:T+H} \in \mathbb{R}^{D \times \delta_T \times (T+H)}$.

The *encoder* Enc($\cdot$) is instantiated as a single linear operator shared across all variates. It operates on the time-delayed sequence $\hat{y}_{1:T+H} \in \mathbb{R}^{D \times \delta_T \times (T+H)}$, where each delay vector of dimension $\delta_T$ is linearly projected to a learnable latent with dimension $M \gg \delta_T$, resulting in the embeddings $z_{1:T+H} \in \mathbb{R}^{D \times M \times (T+H)}$. This lightweight Enc($\cdot$) seeks to preserve the local geometry of the time-delay embedding while enabling expressive comparisons in the latent space used for kernel regression (Eq. 6), which generates the time-delayed prediction $\bar{y}_{T+1:T+H}$.

The *decoder* Dec($\cdot$) maps the time-delayed forecast $\bar{y}_{T+1:T+H} \in \mathbb{R}^{D \times M \times H}$ back to the original time series space. It is implemented using a lightweight MLP shared across all channels. For each channel, the $(M \times H)$ latent matrix is first flattened into a vector, which is then passed

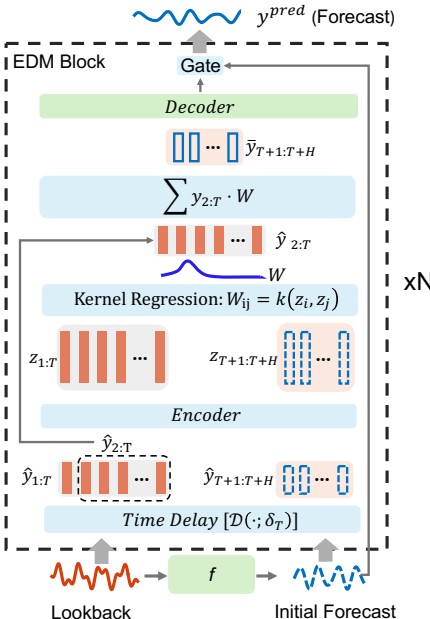

Figure 2. **DeepEDM block** can be stacked, with each subsequent block iteratively refines the prediction of the previous one.

through the MLP to produce the final forecast in $\mathbb{R}^H$, yielding the output $y^{\text{pred}} \in \mathbb{R}^{D \times H}$. Dec($\cdot$) aims to reconstruct the forecast using its noisy time-delayed version.

**DeepEDM block**. We combine the time delay operation, encoder, kernel regression, and decoder into a *DeepEDM block*, as shown in Figure 2. This block receives the historical input $y \in \mathbb{R}^{D \times T}$ in tandem with an initial forecast $y^p \in \mathbb{R}^{D \times H}$ produced by the base predictor, and predicts future time series $y^{\text{pred}} \in \mathbb{R}^{D \times H}$. Importantly, the DeepEDM block is stackable: the output of one block serves as the input forecast to the next, enabling the model to iteratively refine its predictions. To improve gradient flow and training stability, we introduce skip connections from the initial forecast $y^p$ to the final output $y^{\text{pred}}$, modulated by a learnable gating function implemented as a simple linear layer.

**Our full model**. Our DeepEDM model consists of a base predictor $f(\cdot)$, followed by several stacked DeepEDM blocks. Given a lookback window $y$, the base predictor generates a coarse forecast $y^p$, which is successively refined through stacked DeepEDM blocks. All components are differentiable and jointly trained with our loss in Eq. 8.

# 4. Experiments and Results

We evaluate DeepEDM across a wide range of synthetic and real-world benchmarks. Our initial evaluations leverage synthetic datasets derived from well-established nonlinear dynamical systems, allowing us to systematically analyze DeepEDM's capacity to capture complex temporal dependencies. Further, we compare DeepEDM against state-of-the-art deep models on real-world datasets spanning diverse

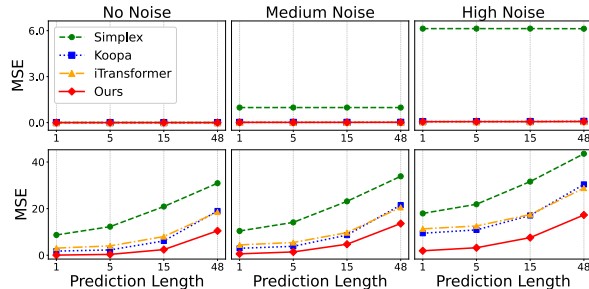

Figure 3. **Results with synthetic data from Lorenz systems**. We plot MSE under varying prediction lengths on non-Chaotic (top) and chaotic (bottom) Lorenz. DeepEDM significantly outperforms baselines in both chaotic and non-chaotic regimes.

domains, including weather, electricity, traffic, and finance. Finally, we provide extensive analysis to assess DeepEDM's ability to generalize to unseen time series, and to study its key design choices. Due to space limits, part of our results, along with extended benchmarks and visualization, are provided in the Appendix.

## 4.1. Experiments on Synthetic Data

We evaluate DeepEDM on synthetic time series generated from (1) non-chaotic Lorenz (Lorenz, 1963), (2) chaotic Lorenz (Lorenz, 1963), and (3) chaotic Rössler (Rössler, 1976) systems. Lorenz and Rössler systems are widely used to study chaotic and non-chaotic dynamics.

**Simulation, setup, and baselines**. To simulate noisy data, we inject Gaussian noise $N(0, \sigma_{noise}^2)$ of various magnitudes of $\sigma_{noise} \in \{0.0, 0.5, 1.0, 1.5, 2.0, 2.5\}$ to the 3 aforementioned systems, resulting in a total of 18 synthetic datasets (see Appendix A.7). We benchmark DeepEDM against three baselines, including EDM with Simplex, Koopa (a deep model integrating Koopman theory), and iTransformer (Transformer-based). Since Simplex is inherently a univariate forecasting method, we run it independently on each variate and aggregate the results to obtain multivariate forecasts. The performance is reported by mean squared error (MSE) and mean absolute error (MAE).

**Results**. Figure 3 shows the forecasting results of all methods across noise levels and prediction horizons. In the low-noise and non-chaotic settings, all methods exhibit comparable performance. However, as noise increases, EDM with *Simplex* degrades sharply, while *DeepEDM* remains robust, achieving lower MSE across all conditions. It also outperforms *Koopa* and *iTransformer* by a small but meaningful margin (see Table 11 in Appendix). In *chaotic* regimes, *DeepEDM*'s advantage is more pronounced, consistently outperforming *Simplex* at all horizons and surpassing *Koopa* and *iTransformer* for longer forecasts. These results underscore *DeepEDM*'s robustness in *noisy, chaotic* environments while exceeding both classical EDM and modern baselines. Additional results, including those for the Rössler system,

*Table 1.* **Multivariate forecasting results** with different forecast lengths $H \in \{24, 36, 48, 60\}$ for ILI and $H \in \{48, 96, 144, 192\}$ for others. We set the lookback length $T = 2H$. **Bold** indicates the best performance, while 2nd best is underlined. In case of a draw, both models are considered winners. Gray represents dynamical systems. *Source: When available, results are taken directly from (Liu et al., 2024b); otherwise reproduced using their official code run with reported metrics averaged over 5 runs with different random seeds.*

| Models | | **Ours** | | Koopa | | KNF | | Attraos | | CycleNet | | iTransformer | | PatchTST | | TimeMixer | | DLinear | | FITS | | MICN | | Naïve | |
|---|---|---|---|---|---|---|---|---|---|---|---|---|---|---|---|---|---|---|---|---|---|---|---|---|---|
| Metric | | MSE | MAE | MSE | MAE | MSE | MAE | MSE | MAE | MSE | MAE | MSE | MAE | MSE | MAE | MSE | MAE | MSE | MAE | MSE | MAE | MSE | MAE | MSE | MAE |
| ETTh1 | 48 | **0.324** | **0.357** | 0.336 | 0.377 | 0.876 | 0.709 | 0.341 | 0.371 | 0.331 | 0.370 | 0.343 | 0.380 | 0.337 | 0.375 | 0.336 | 0.375 | 0.343 | 0.371 | 0.344 | 0.370 | 0.375 | 0.406 | 1.268 | 0.695 |
| | 96 | **0.365** | **0.384** | 0.371 | 0.405 | 0.975 | 0.744 | 0.387 | 0.402 | 0.389 | 0.404 | 0.392 | 0.411 | 0.372 | 0.393 | 0.388 | 0.405 | 0.379 | 0.393 | 0.381 | 0.395 | 0.406 | 0.429 | 1.294 | 0.713 |
| | 144 | **0.388** | **0.398** | 0.405 | 0.418 | 0.801 | 0.662 | 0.415 | 0.422 | 0.415 | 0.422 | 0.424 | 0.430 | 0.394 | 0.412 | 0.413 | 0.421 | 0.393 | 0.403 | 0.396 | 0.406 | 0.437 | 0.448 | 1.316 | 0.725 |
| | 192 | 0.407 | 0.421 | 0.416 | 0.429 | 0.941 | 0.744 | 0.429 | 0.434 | 0.433 | 0.436 | 0.446 | 0.449 | 0.416 | 0.439 | 0.443 | 0.447 | 0.407 | 0.416 | **0.405** | **0.414** | 0.518 | 0.496 | 1.325 | 0.733 |
| ETTh2 | 48 | 0.225 | **0.288** | 0.226 | 0.300 | 0.385 | 0.376 | 0.230 | 0.301 | 0.238 | 0.305 | 0.243 | 0.314 | **0.223** | 0.297 | 0.230 | 0.302 | 0.226 | 0.305 | 0.227 | 0.298 | 0.260 | 0.336 | 0.344 | 0.374 |
| | 96 | 0.289 | **0.333** | 0.297 | 0.349 | 0.433 | 0.446 | 0.302 | 0.350 | 0.306 | 0.357 | 0.302 | 0.356 | 0.300 | 0.353 | 0.298 | 0.350 | 0.294 | 0.351 | **0.287** | 0.341 | 0.343 | 0.393 | 0.432 | 0.422 |
| | 144 | 0.324 | **0.362** | 0.333 | 0.381 | 0.441 | 0.456 | 0.355 | 0.383 | 0.350 | 0.388 | 0.346 | 0.386 | 0.346 | 0.390 | 0.339 | 0.383 | 0.354 | 0.397 | **0.315** | 0.363 | 0.374 | 0.411 | 0.484 | 0.448 |
| | 192 | 0.351 | 0.377 | 0.356 | 0.393 | 0.528 | 0.503 | 0.373 | 0.399 | 0.377 | 0.407 | 0.383 | 0.409 | 0.383 | 0.406 | 0.359 | 0.406 | 0.385 | 0.418 | **0.334** | **0.376** | 0.455 | 0.464 | 0.534 | 0.472 |
| ETTm1 | 48 | **0.277** | **0.318** | 0.283 | 0.333 | 1.026 | 0.792 | 0.312 | 0.353 | 0.283 | 0.336 | 0.314 | 0.358 | 0.286 | 0.336 | 0.302 | 0.349 | 0.322 | 0.355 | 0.324 | 0.357 | 0.294 | 0.353 | 1.165 | 0.638 |
| | 96 | **0.288** | **0.328** | 0.294 | 0.345 | 0.957 | 0.782 | 0.314 | 0.355 | 0.302 | 0.353 | 0.304 | 0.354 | 0.299 | 0.346 | 0.299 | 0.348 | 0.309 | 0.346 | 0.310 | 0.346 | 0.306 | 0.364 | 1.214 | 0.665 |
| | 144 | **0.308** | **0.344** | 0.322 | 0.366 | 0.921 | 0.760 | 0.332 | 0.368 | 0.327 | 0.368 | 0.331 | 0.373 | 0.325 | 0.363 | 0.326 | 0.365 | 0.327 | 0.359 | 0.326 | 0.358 | 0.342 | 0.390 | 1.246 | 0.682 |
| | 192 | **0.322** | **0.353** | 0.337 | 0.378 | 0.896 | 0.731 | 0.349 | 0.378 | 0.346 | 0.382 | 0.345 | 0.383 | 0.343 | 0.375 | 0.345 | 0.378 | 0.337 | 0.365 | 0.338 | 0.365 | 0.386 | 0.415 | 1.261 | 0.690 |
| ETTm2 | 48 | 0.133 | 0.221 | 0.134 | 0.226 | 0.621 | 0.623 | 0.139 | 0.236 | **0.123** | **0.216** | 0.139 | 0.234 | 0.135 | 0.231 | 0.136 | 0.229 | 0.144 | 0.240 | 0.145 | 0.242 | 0.131 | 0.238 | 0.220 | 0.295 |
| | 96 | 0.169 | **0.248** | 0.171 | 0.254 | 1.535 | 1.012 | 0.174 | 0.259 | **0.164** | 0.249 | 0.181 | 0.269 | 0.171 | 0.255 | 0.174 | 0.257 | 0.172 | 0.256 | 0.172 | 0.257 | 0.197 | 0.295 | 0.267 | 0.328 |
| | 144 | 0.203 | **0.271** | 0.206 | 0.280 | 1.337 | 0.876 | 0.209 | 0.284 | 0.212 | 0.286 | 0.214 | 0.294 | 0.205 | 0.282 | 0.207 | 0.284 | **0.200** | 0.276 | 0.200 | 0.277 | 0.210 | 0.297 | 0.307 | 0.352 |
| | 192 | 0.224 | **0.289** | 0.226 | 0.298 | 1.355 | 0.908 | 0.233 | 0.302 | 0.231 | 0.302 | 0.238 | 0.310 | 0.221 | 0.294 | 0.229 | 0.297 | **0.219** | 0.290 | 0.220 | 0.291 | 0.248 | 0.328 | 0.340 | 0.371 |
| ECL | 48 | 0.161 | 0.247 | 0.130 | 0.234 | 0.175 | 0.265 | 0.192 | 0.268 | **0.120** | **0.215** | 0.134 | 0.226 | 0.147 | 0.246 | 0.142 | 0.235 | 0.158 | 0.241 | 0.203 | 0.279 | 0.156 | 0.271 | 1.543 | 0.925 |
| | 96 | 0.137 | 0.232 | 0.136 | 0.236 | 0.198 | 0.284 | 0.150 | 0.244 | **0.127** | **0.222** | 0.134 | 0.230 | 0.143 | 0.241 | 0.134 | 0.227 | 0.153 | 0.245 | 0.154 | 0.248 | 0.165 | 0.277 | 1.588 | 0.946 |
| | 144 | 0.145 | 0.239 | 0.149 | 0.247 | 0.204 | 0.297 | 0.151 | 0.246 | **0.138** | **0.232** | 0.146 | 0.240 | 0.145 | 0.241 | 0.145 | 0.235 | 0.152 | 0.245 | 0.152 | 0.246 | 0.163 | 0.274 | 1.605 | 0.953 |
| | 192 | 0.151 | 0.244 | 0.156 | 0.254 | 0.245 | 0.321 | 0.154 | 0.249 | **0.146** | **0.241** | 0.155 | 0.249 | 0.147 | **0.240** | 0.163 | 0.255 | 0.153 | 0.246 | 0.154 | 0.247 | 0.171 | 0.284 | 1.596 | 0.951 |
| Exchange | 48 | **0.042** | 0.142 | **0.042** | 0.143 | 0.128 | 0.271 | 0.045 | 0.147 | 0.044 | 0.144 | 0.045 | 0.148 | 0.044 | 0.144 | 0.043 | 0.143 | 0.043 | 0.145 | 0.054 | 0.180 | 0.117 | 0.248 | **0.042** | **0.139** |
| | 96 | 0.088 | 0.205 | 0.083 | 0.207 | 0.294 | 0.394 | 0.093 | 0.213 | 0.089 | 0.209 | 0.095 | 0.219 | 0.085 | 0.204 | 0.084 | 0.203 | 0.084 | 0.220 | 0.113 | 0.261 | 0.108 | 0.251 | **0.081** | **0.196** |
| | 144 | 0.133 | 0.255 | 0.130 | 0.261 | 0.597 | 0.578 | 0.151 | 0.274 | 0.144 | 0.267 | 0.154 | 0.283 | 0.132 | 0.260 | 0.146 | 0.270 | 0.132 | 0.253 | 0.133 | 0.258 | 0.152 | 0.301 | **0.122** | **0.244** |
| | 192 | 0.178 | 0.301 | 0.184 | 0.309 | 0.654 | 0.595 | 0.205 | 0.323 | 0.207 | 0.322 | 0.212 | 0.334 | 0.174 | 0.300 | 0.196 | 0.316 | 0.178 | 0.299 | 0.182 | 0.305 | 0.187 | 0.331 | **0.167** | **0.289** |
| Traffic | 48 | 0.448 | 0.286 | 0.415 | 0.274 | 0.621 | 0.382 | 0.612 | 0.396 | 0.437 | 0.290 | **0.369** | **0.257** | 0.426 | 0.286 | 0.445 | 0.283 | 0.488 | 0.352 | 0.704 | 0.419 | 0.496 | 0.301 | 2.641 | 1.057 |
| | 96 | 0.383 | 0.259 | 0.401 | 0.275 | 0.645 | 0.376 | 0.439 | 0.300 | 0.406 | 0.276 | **0.365** | **0.259** | 0.413 | 0.283 | 0.406 | 0.277 | 0.485 | 0.336 | 0.457 | 0.306 | 0.511 | 0.312 | 2.715 | 1.077 |
| | 144 | 0.380 | 0.258 | 0.397 | 0.276 | 0.683 | 0.402 | 0.423 | 0.294 | 0.402 | 0.275 | **0.373** | 0.266 | 0.405 | 0.278 | 0.391 | 0.263 | 0.452 | 0.317 | 0.432 | 0.293 | 0.498 | 0.309 | 2.739 | 1.084 |
| | 192 | 0.387 | **0.262** | 0.403 | 0.284 | 0.699 | 0.405 | 0.421 | 0.295 | 0.402 | 0.275 | **0.374** | 0.267 | 0.404 | 0.277 | 0.424 | 0.293 | 0.438 | 0.309 | 1.313 | 0.776 | 0.494 | 0.312 | 2.747 | 1.085 |
| Weather | 48 | 0.138 | **0.168** | 0.126 | 0.168 | 0.201 | 0.288 | 0.149 | 0.191 | 0.129 | 0.171 | 0.137 | 0.174 | 0.140 | 0.179 | 0.131 | 0.174 | 0.156 | 0.198 | 0.157 | 0.200 | 0.157 | 0.217 | 0.194 | 0.193 |
| | 96 | 0.157 | **0.192** | 0.154 | 0.205 | 0.295 | 0.308 | 0.168 | 0.214 | 0.155 | 0.203 | 0.169 | 0.215 | 0.160 | 0.206 | 0.155 | 0.205 | 0.186 | 0.229 | 0.187 | 0.231 | 0.187 | 0.250 | 0.259 | 0.254 |
| | 144 | 0.174 | **0.210** | 0.172 | 0.225 | 0.394 | 0.401 | 0.184 | 0.231 | **0.171** | 0.223 | 0.187 | 0.234 | 0.174 | 0.221 | 0.173 | 0.223 | 0.199 | 0.244 | 0.199 | 0.244 | 0.197 | 0.257 | 0.284 | 0.274 |
| | 192 | **0.191** | **0.226** | 0.193 | 0.241 | 0.462 | 0.437 | 0.202 | 0.249 | 0.192 | 0.243 | 0.206 | 0.253 | 0.195 | 0.243 | 0.193 | 0.243 | 0.217 | 0.261 | 0.217 | 0.261 | 0.214 | 0.270 | 0.309 | 0.292 |
| ILI | 24 | 1.799 | **0.797** | 1.621 | 0.800 | 3.722 | 1.432 | - | - | 2.188 | 0.940 | 1.966 | 0.888 | 2.063 | 0.881 | 2.147 | 0.899 | 2.624 | 1.118 | 3.311 | 1.311 | 4.380 | 1.558 | 6.213 | 1.622 |
| | 36 | **1.655** | **0.768** | 1.803 | 0.855 | 3.941 | 1.448 | - | - | 2.113 | 0.949 | 1.827 | 0.865 | 2.178 | 0.943 | 1.892 | 0.894 | 2.693 | 1.156 | 3.112 | 1.232 | 3.314 | 1.313 | 7.714 | 1.906 |
| | 48 | **1.616** | 0.789 | 1.768 | 0.903 | 3.287 | 1.377 | 2.437 | 1.084 | 1.849 | 0.919 | 1.748 | 0.908 | 1.916 | 0.896 | 1.874 | 0.915 | 2.852 | 1.229 | 3.156 | 1.290 | 2.457 | 1.085 | 7.851 | 1.952 |
| | 60 | 1.719 | **0.831** | 1.743 | 0.891 | 2.974 | 1.301 | 2.341 | 1.064 | 1.872 | 0.932 | 2.077 | 0.999 | 1.981 | 0.917 | 2.187 | 0.991 | 2.554 | 1.144 | 3.337 | 1.280 | 2.379 | 1.040 | 6.885 | 1.788 |
| 1st Count | | 36 | | 5 | | 0 | | 0 | | 11 | | 6 | | 2 | | 0 | | 2 | | 7 | | 0 | | 8 | |

Note: The official code of Attraos (Hu et al., 2024) does not support $H = \{24, 36\}$ for ILI dataset. Therefore, we report these entries as empty rather than extensively modifying their code to make it work.

are provided in Appendix A.7.

### 4.2. Experiments on Forecasting Benchmarks

Moving forward, we conduct comprehensive evaluations on standard time series forecasting benchmarks.

**Datasets**. We consider both multivariate and univariate time series forecasting. For multivariate forecasting, we evaluate on 10 real-world datasets: *ETTh1*, *ETTh2*, *ETTm1*, *ETTm2* (Zhou et al., 2021), National Illness (*ILI*) (Lai et al., 2018), *Solar-Energy* (Lai et al., 2018) (see appendix), *Electricity* (see appendix), *Traffic* (PeMS) (Wu et al., 2021), *Weather* (Wetterstation) (Wu et al., 2021), and *Exchange* (Lai et al., 2018). For univariate forecasting, we leverage the well-established *M4* dataset (Makridakis et al., 2020) (see appendix), which contains 6 subsets of periodically collected univariate marketing data. These datasets encompass different domains and exhibit diverse temporal

patterns, allowing for a robust assessment.

**Setup**. Our experimental protocol adheres to the pre-processing methods and data split ratios established by prominent prior works such as TimesNet (Wu et al., 2023) and Koopa (Liu et al., 2024b). For all experiments, we use the Time-Series-Library (Wang et al., 2024b) to ensure consistency and comparability. For our main results, we adopt the adaptive lookback windowing approach from Koopa (Liu et al., 2024b), where the lookback window length $T$ is set to twice the forecast horizon $H$. We also report results with lookback window search in the appendix.

**Baselines**. We consider a set of strong baselines. While emphasizing comparisons with dynamical system-based methods such as *Koopa* (Liu et al., 2024b), *Attraos* (Hu et al., 2024) and *KNF* (Wang et al., 2023b), we also include other popular baselines. These include MLP-based models like *TimeMixer*, *FITS*, and *DLinear*, as well as Transformer-based models such as *iTransformer* and *PatchTST*. Additionally, we also benchmark the *Naïve* baseline as described by (Hewamalage et al., 2023) (i.e. predicting the last value of lookback window as forecast) to provide the simplest benchmark to assess relative performance.

**Results**. Our main results are summarized in Table 1 (see variance in Appendix Table 10). *DeepEDM* achieves state-of-the-art performance on the multivariate forecasting benchmarks, winning on *36* metrics compared to *5* for the next-best dynamical system-based method, *Koopa*, and *11* for the strongest deep learning model, *CycleNet* (Lin et al., 2024). These results highlight DeepEDM's effectiveness and versatility across diverse domains. Notably, *DeepEDM* excels at the MAE metric, which is less sensitive to outliers, suggesting a stronger ability to capture underlying trends. Interestingly, the *Naïve* baseline, outperforms all models in case of *Exchange* (Stocks) dataset, consistent with findings of (Hewage et al., 2020), thus revealing the blind spots of many forecasting models. Beyond multivariate settings, *DeepEDM* also exhibits strong performance in univariate forecasting on the *M4* dataset (see Appendix A.3).

### 4.3. Further Analyses

#### Generalization to Unseen Time Series

**Rationale**. Time series forecasting benchmarks typically employ temporal splits for evaluation, that is, training on earlier time steps and testing on later ones. To evaluate the generalization across sequences, we considers a more challenging setting: splitting across different time series (i.e., channels) within the same dataset.

**Setup and datasets**. In addition to the standard temporal train-test split, we also partition the time series (variates) in ETT, Exchange, and Weather into disjoint training and testing sets, ensuring no overlap in sequence identity. For

*Table 2.* **Generalization to unseen time series**. Each model is trained on a subset of sequences and evaluated on disjoint, unseen sequences from the same dataset. DeepEDM achieves the best MAE and MSE in 39 out of 48 settings.

| Models | | **Ours** | | Koopa | | PatchTST | | iTransformer | |
|---|---|---|---|---|---|---|---|---|---|
| | | MSE | MAE | MSE | MAE | MSE | MAE | MSE | MAE |
| ETTh1 | 48 | **0.2182** | **0.2980** | 0.2383 | 0.3190 | 0.2312 | 0.3090 | 0.2474 | 0.3190 |
| | 96 | **0.2230** | **0.3120** | 0.2382 | 0.3270 | 0.2601 | 0.3370 | 0.2535 | 0.3340 |
| | 144 | **0.2285** | **0.3190** | 0.2598 | 0.3420 | 0.2567 | 0.3380 | 0.2634 | 0.3420 |
| | 192 | **0.2510** | **0.3400** | 0.2555 | 0.3420 | 0.2637 | 0.3430 | 0.2702 | 0.3480 |
| ETTh2 | 48 | **0.0931** | **0.1850** | 0.1181 | 0.2160 | 0.1114 | 0.2090 | 0.1107 | 0.2110 |
| | 96 | **0.1377** | **0.2260** | 0.1517 | 0.2440 | 0.1653 | 0.2590 | 0.1555 | 0.2510 |
| | 144 | **0.1795** | **0.2640** | 0.1979 | 0.2850 | 0.2265 | 0.2980 | 0.1914 | 0.2760 |
| | 192 | **0.1956** | **0.2770** | 0.2115 | 0.2950 | 0.2297 | 0.3170 | 0.2209 | 0.2970 |
| ETTm1 | 48 | **0.2068** | **0.2680** | 0.2377 | 0.2960 | 0.2208 | 0.2830 | 0.2305 | 0.2910 |
| | 96 | 0.2141 | **0.2780** | 0.2337 | 0.3020 | **0.2090** | 0.2840 | 0.2386 | 0.3020 |
| | 144 | 0.2142 | **0.2880** | 0.2518 | 0.3180 | **0.2134** | 0.2960 | 0.2590 | 0.3180 |
| | 192 | **0.2194** | **0.2980** | 0.2454 | 0.3200 | 0.2212 | 0.3070 | 0.2512 | 0.3200 |
| ETTm2 | 48 | **0.0544** | **0.1470** | 0.0641 | 0.1640 | 0.0710 | 0.1740 | 0.0892 | 0.1950 |
| | 96 | **0.0659** | **0.1590** | 0.0814 | 0.1820 | 0.0768 | 0.1750 | 0.0861 | 0.1890 |
| | 144 | **0.0784** | **0.1710** | 0.0920 | 0.1910 | 0.0917 | 0.1880 | 0.1004 | 0.1990 |
| | 192 | **0.1024** | **0.1940** | 0.1073 | 0.2060 | 0.1115 | 0.2050 | 0.1074 | 0.2070 |
| Exchange | 48 | **0.0388** | **0.1290** | 0.0459 | 0.1420 | 0.0431 | 0.1370 | 0.0428 | 0.1390 |
| | 96 | **0.0783** | **0.1860** | 0.0936 | 0.2120 | 0.0828 | 0.1930 | 0.0912 | 0.2030 |
| | 144 | 0.1330 | **0.2390** | 0.1725 | 0.2820 | **0.1279** | 0.2490 | 0.1772 | 0.2800 |
| | 192 | 0.1754 | **0.2780** | 0.2706 | 0.3560 | **0.1667** | 0.2840 | 0.2136 | 0.3140 |
| Weather | 48 | **0.2915** | 0.2440 | 0.3479 | 0.2780 | 0.3030 | **0.2410** | 0.3967 | 0.2870 |
| | 96 | 0.2977 | **0.2510** | 0.3142 | 0.2760 | **0.2934** | 0.2620 | 0.4462 | 0.3260 |
| | 144 | 0.2928 | **0.2520** | 0.3101 | 0.2820 | **0.2841** | 0.2570 | 0.4085 | 0.3220 |
| | 192 | 0.2917 | 0.2630 | 0.3261 | 0.2900 | **0.2907** | **0.2620** | 0.4393 | 0.3430 |

the ETT datasets, which contain 7 sequences, we train on sequences 0–2 using only timesteps from the standard training split, and test on sequences 4–6 using the standard test split. This 3 : 3 split is necessary as several baseline models are unable to handle differing input dimensions between training and testing. Similarly, for the Exchange dataset (8 sequences), we train on the first 4 sequences and test on the last 4. For the Weather dataset (21 sequences), we train on sequences 0–9 and test on sequences 10–19.

**Baselines**. We compare DeepEDM to three representative baselines, including Koopa, iTransformer, and PatchTST. The forecasting horizon $H$ varies over $\{48, 96, 144, 192\}$, with the lookback window set to $2H$ in all cases.

**Results**. Table 2 shows the results. DeepEDM leads the performance in both MAE and MSE, ranking first in 39 out of 48 settings. The results demonstrate DeepEDM's ability to generalize across different time series.

#### Robustness to Measurement Noise

**Rationale**. We hypothesize that DeepEDM's learned latent space functions as a noise-robust kernel that more accurately preserves the local neighborhood structure of the underlying state space than time-delay embeddings. We conduct experiments with simulated date to verify this hypothesis.

**Simulation and setup**. We simulate trajectories using a chaotic Lorenz system with $\sigma = 10.0$, $\rho = 28.0$, $\beta = 2.667$, and initial conditions: $(0.0, 1.0, 1.05)$, where the ground-truth states $\mathbf{x}_t \in \mathbb{R}^3$ are known. To satisfy the univariate embedding requirement of Takens' theorem, we consider a

*Table 3.* **Model design ablation**. We evaluate the effects of progressively incorporating key components into our model, with metrics averaged over *four prediction lengths and three random seeds* ($\pm\sigma$ shown). Each successive addition yields consistent improvements across most metrics *relative to the preceding configuration*.

| Dataset | Linear | | MLP | | MLP+EDM | | Full Model | |
|---|---|---|---|---|---|---|---|---|
| | *MSE* | *MAE* | *MSE* | *MAE* | *MSE* | *MAE* | *MSE* | *MAE* |
| ECL | $0.1646_{\pm0.0001}$ | $0.2528_{\pm0.0001}$ | $0.1616_{\pm0.0005}$ | $0.2532_{\pm0.0004}$ | $0.1491_{\pm0.0003}$ | $0.2409_{\pm0.0001}$ | $\mathbf{0.1487}_{\pm0.0003}$ | $\mathbf{0.2404}_{\pm0.0004}$ |
| ETTh1 | $0.3805_{\pm0.0004}$ | $0.3907_{\pm0.0003}$ | $0.3782_{\pm0.0000}$ | $0.3915_{\pm0.0002}$ | $0.3782_{\pm0.0010}$ | $0.3939_{\pm0.0002}$ | $\mathbf{0.3702}_{\pm0.0033}$ | $\mathbf{0.3897}_{\pm0.0020}$ |
| ETTh2 | $0.2951_{\pm0.0004}$ | $0.3403_{\pm0.0003}$ | $\mathbf{0.2910}_{\pm0.0010}$ | $\mathbf{0.3377}_{\pm0.0005}$ | $0.3017_{\pm0.0039}$ | $0.3433_{\pm0.0019}$ | $0.2954_{\pm0.0031}$ | $0.3391_{\pm0.0014}$ |
| ETTm1 | $0.3168_{\pm0.0004}$ | $0.3449_{\pm0.0004}$ | $0.3123_{\pm0.0002}$ | $0.3442_{\pm0.0001}$ | $0.3019_{\pm0.0003}$ | $0.3378_{\pm0.0002}$ | $\mathbf{0.2984}_{\pm0.0004}$ | $\mathbf{0.3357}_{\pm0.0003}$ |
| ETTm2 | $0.1838_{\pm0.0002}$ | $0.2602_{\pm0.0001}$ | $0.1836_{\pm0.0002}$ | $0.2598_{\pm0.0001}$ | $0.1830_{\pm0.0011}$ | $0.2585_{\pm0.0008}$ | $\mathbf{0.1817}_{\pm0.0015}$ | $\mathbf{0.2572}_{\pm0.0008}$ |
| Traffic | $0.5001_{\pm0.0001}$ | $0.3226_{\pm0.0007}$ | $0.4521_{\pm0.0016}$ | $0.3104_{\pm0.0014}$ | $\mathbf{0.3930}_{\pm0.0023}$ | $0.2683_{\pm0.0015}$ | $0.4001_{\pm0.0004}$ | $\mathbf{0.2663}_{\pm0.0001}$ |
| Exchange | $0.1097_{\pm0.0004}$ | $\mathbf{0.2247}_{\pm0.0005}$ | $0.1110_{\pm0.0015}$ | $0.2262_{\pm0.0011}$ | $0.1122_{\pm0.0023}$ | $0.2278_{\pm0.0019}$ | $\mathbf{0.1090}_{\pm0.0011}$ | $\mathbf{0.2247}_{\pm0.0015}$ |
| ILI | $2.0240_{\pm0.0540}$ | $0.9271_{\pm0.0163}$ | $1.9827_{\pm0.0762}$ | $0.8864_{\pm0.0271}$ | $1.6857_{\pm0.0311}$ | $0.7977_{\pm0.0091}$ | $\mathbf{1.6779}_{\pm0.0391}$ | $\mathbf{0.7935}_{\pm0.0087}$ |
| Weather | $0.1955_{\pm0.0003}$ | $0.2249_{\pm0.0011}$ | $0.1899_{\pm0.0002}$ | $0.2198_{\pm0.0003}$ | $0.1660_{\pm0.0007}$ | $0.2002_{\pm0.0007}$ | $\mathbf{0.1651}_{\pm0.0004}$ | $\mathbf{0.1989}_{\pm0.0004}$ |
| #Improvements | Baseline | | 14 | | 13 | | 17 | |
| #Degradations | | | 4 | | 5 | | 1 | |

*Table 4.* **Robustness to noise**. We compare time-delayed embeddings with our learned kernel for $K$-nearest neighbor retrieval on simulated data, reporting mean recall as the evaluation metric.

| | $\delta_T$ | Recall (clean) | | Recall (noisy) | |
|---|---|---|---|---|---|
| | | Time-delayed | Learned (ours) | Time-delayed | Learned (ours) |
| $K$=1 | 1 | 0.707 | **0.990** | 0.082 | **0.849** |
| | 5 | 0.986 | **0.990** | 0.257 | **0.957** |
| | 10 | **0.998** | 0.990 | 0.396 | **0.973** |
| $K$=7 | 7 | 0.545 | **0.586** | 0.220 | **0.527** |
| | 14 | 0.728 | **0.753** | 0.368 | **0.622** |
| | 28 | **0.896** | 0.857 | 0.564 | **0.730** |

measurement function that takes a single dimension of $\mathbf{x}_t$ as $y_t$. All experiments are run independently for each of the 3 dimensions, and the averaged metrics are reported. For each time step $t$, we identify $K$ nearest neighbors in the state space using Euclidean distance, and treat them as the ground-truth. We then retrieve top $K$ neighbors using: (i) time-delay embeddings of $y_t$, or (ii) via distances computed with the learned kernel in Eq. 6. We compare the retrieved neighbors against the ground-truth, and report mean recall. This is done across $K(\in [1, 7])$ and similar to Section 4.1 under two noise settings: (i) noise-free, and (ii) with additive Gaussian noise ($\sigma_{noise} = 2.5$).

**Results**. As shown in Table 4, both methods achieve high recall under noise-free conditions. However, when noise is introduced, recall for the time-delay embedding drops sharply. In contrast, our learned kernel degrades more gracefully, maintaining significantly higher recall. This suggests that the latent space preserves the topological structure of the true state space more effectively in the presence of noise. Our DeepEDM thus offers robustness to input noise, providing a key advantage over EDM in real-world applications.

**Model Design Ablation**

**Rationale**. We conduct an ablation study to evaluate the contribution of each component in DeepEDM.

**Setup and datasets**. We begin with a minimal baseline consisting of a single linear layer, and incrementally add: (1) a multilayer perceptron (MLP) (w. MSE loss), (2) EDM blocks (w. MSE loss), and finally (3) the full model (w. optimized loss). This ablation allows us to isolate and quantify the impact of each component on overall forecasting performance. All ablation experiments are conducted on 9 standard multivariate time series benchmark datasets, using 4 different prediction lengths per dataset. Each setting is repeated with 3 random seeds. We report MSE and MAE, averaged across both prediction lengths and seeds, to ensure statistically robust and comprehensive evaluation. Additional ablations on the effects of the $\mathcal{L}_{td}$ loss, as well as the choice of lookback length, time delay, and embedding dimensions, can be found in Appendix A.5.

**Results**. Table 3 summarizes the main ablation results (full results in Appendix Table 12 and Table 9). The simple linear baseline performs the worst, confirming the inadequacy of linear models for nonlinear temporal dynamics. Introducing an MLP leads to a moderate improvement, while the inclusion of EDM blocks yields significant gains across most datasets—demonstrating their effectiveness in capturing nonlinear and multiscale interactions. Incorporating optimized loss further refines performance, indicating the benefit of aligning the optimization objective with dynamical structure. Our results provide clear empirical support for each design choice in DeepEDM.

## 5. Conclusion

In this paper, we presented *DeepEDM*, a novel framework that integrates dynamical systems modeling and deep neural networks for time series forecasting. By leveraging time-delayed embeddings and kernel regression in a latent space, DeepEDM effectively captures underlying dynamics with noisy input, delivering state-of-the-art performance across synthetic and real-world benchmarks. Future work should explore the more advanced S-map (Chang et al., 2017) method within EDM, for even greater flexibility in modeling nonlinear dynamics.

## Acknowledgment

This work was partially supported by National Science Foundation under Grant No. CNS 2333491, and by the Army Research Lab under contract number W911NF-2020221.

## Impact Statement

This work presents a novel approach to time series forecasting. The potential broader impact includes improved forecasting accuracy in various domains such as economics, energy, transportation, and meteorology, leading to better decision-making and resource allocation. However, it is important to acknowledge that improved forecasting accuracy may also lead to unintended consequences, such as over-reliance on predictions or misuse of predictive models. It is crucial to use forecasting tools responsibly and ethically, considering potential biases in data and models, and ensuring transparency and accountability in their applications.

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

# A. Appendix

This appendix provides additional details on several aspects of our study. First, we provide more details on the terminology essential for the background on our method (A.1). Next, we outline the details of the implementation, optimization, and training of *DeepEDM* (A.2). Further, we describe experiments conducted on the short-term forecasting M4 benchmark (A.3) and results from the standard lookback searching setting for long-term forecasting (A.4). Additionally, we include detailed results of experiments studying the impact of lookback length (A.5.1), sensitivity to time delay and embedding dimension (A.5.2), loss function (A.5.3) and stability of our results (A.6). Finally, we elaborate on the synthetic data experiments (A.7).

## A.1. Terminology and Definitions

**Definition A.1** (Manifold). A *manifold $M$* of dimension $d$ is a topological space that is locally homeomorphic to $\mathbb{R}^d$, which means that every point in $M$ has a neighborhood that resembles an open subset of $\mathbb{R}^d$. If $M$ has a smooth structure, allowing for differentiation, it is called a *smooth manifold*.

**Definition A.2** (Smooth Map). A function $f : M \to N$ between smooth manifolds is called *smooth* if it has continuous derivatives of all orders in local coordinates.

**Definition A.3** (Homeomorphism). A function $f : X \to Y$ between topological spaces is a *homeomorphism* if it is a continuous bijection with a continuous inverse. This ensures that $X$ and $Y$ have the same topological structure.

**Definition A.4** (Diffeomorphism). A *diffeomorphism* is a smooth function $f : M \to N$ between smooth manifolds that is bijective and has a smooth inverse. If such a map exists, $M$ and $N$ are said to be *diffeomorphic*, meaning they have the same smooth structure.

**Definition A.5** (Immersion). A smooth map $f : M \to N$ is an *immersion* if its differential $df_p : T_pM \to T_{f(p)}N$ is injective at every point $p \in M$. If $f$ is also injective as a function, it is called an *injective immersion*.

**Definition A.6** (Submanifold). A subset $S \subset \mathbb{R}^m$ is a *submanifold* if it is a manifold itself and the inclusion map $i : S \hookrightarrow \mathbb{R}^m$ is an embedding. This means that $S$ locally resembles a lower-dimensional Euclidean space and inherits a smooth structure from $\mathbb{R}^m$.

**Definition A.7** (Embedding). An *embedding* of a smooth manifold $M$ into $\mathbb{R}^m$ is a smooth injective immersion that is also a homeomorphism onto its image. This means that the map preserves both the local differential structure and the topology of $M$, ensuring that $M$ is faithfully represented in $\mathbb{R}^m$ without self-intersections or distortions.

**Definition A.8** (Generic Choice). A property is said to hold for a *generic choice* of a parameter (such as the delay $\tau$ or observation function $h$) if it holds for all choices in a residual subset of the parameter space. Residual sets are dense in the appropriate function space and contain a countable intersection of open dense sets, meaning that "almost every" choice satisfies the property in a topological sense.

## A.2. Implementation Details

**Implementation:** Similar to most of the baselines, we developed *DeepEDM* within the popular *Time-Series-Library* benchmarking repository (Wu et al., 2023; Wang et al., 2024b) to ensure methodological consistency with baseline approaches in data preprocessing, splitting, and evaluation metrics (MSE and MAE). We also ensure our implementation is free of the "**drop last**" bug as reported by (Qiu et al., 2024) that can artificially inflate evaluation metrics.

In our implementation, the base predictor $f$ is instantiated as an MLP with 1 to 3 layers, each followed by a non-linear activation and dropout. The number of *DeepEDM* blocks is also varied between 1 and 3 based on dataset size, with larger datasets benefiting from increased expressivity through deeper architectures.

**Normalization:** Following prior work (Li et al., 2023; Liu et al., 2024b; Nie et al., 2023), we also apply reversible instance normalization (Kim et al., 2022) to the input history and output predictions.

**Loss function:** The primary optimization objective for the *DeepEDM* model is minimizing the error between the predicted forecast and true forecast mathematically formalized as:

$$\mathcal{L}_{\text{err}} = \frac{1}{H} \sum_{i=1}^{H} ||y_i - y_i^{\text{pred}}|| \tag{9}$$

where $y_i$ signifies the actual value and $y_i^{\text{pred}}$ represents the value predicted by the model at timestep $i$. For the long-term

forecasting tasks, we follow (Xiong et al., 2024) in optimizing the first temporal difference errors ($\mathcal{L}_{\mathrm{td}}$) defined as:

$$\mathcal{L}_{\mathrm{td}} = \frac{1}{H} \sum_{i=1}^{H} \ell(\nabla y_{t+i}, \nabla y_{t+i}^{\mathrm{pred}}) \tag{10}$$

Here, $\nabla y_{t+i}$ and $\nabla y_{t+i}^{\mathrm{pred}}$ denote the true and predicted first differences (i.e. $y_{t+1} - y_t$ and $y_{t+1}^{\mathrm{pred}} - y_t^{\mathrm{pred}}$)), respectively. The function $\ell$ evaluates the mean absolute error for these differences, thus focusing on the accuracy of sequential changes of the series. Further following the methodology proposed by (Xiong et al., 2024), we also consider the balance between these loss components using $\lambda$, defined as:

$$\lambda = \frac{1}{H} \sum_{i=1}^{H} \mathbf{1}(\mathrm{sgn}(\nabla y_{t+i}) \neq \mathrm{sgn}(\nabla y_{t+i}^{\mathrm{pred}})) \tag{11}$$

Here *sgn* refers to the Signum function. The final composite loss function $L$ is then computed as a weighted sum of $\mathcal{L}_{\mathrm{err}}$ and $\mathcal{L}_{\mathrm{td}}$, modulated by $\lambda$:

$$\mathcal{L} = \lambda \cdot \mathcal{L}_{\mathrm{err}} + (1 - \lambda) \cdot \mathcal{L}_{\mathrm{td}} \tag{12}$$

The parameter $\lambda$ dynamically adjusts the weighting between the $\mathcal{L}_{\mathrm{err}}$ and $\mathcal{L}_{\mathrm{td}}$ based on the frequency of sign changes between the actual and predicted differences, promoting higher fidelity in capturing dynamic temporal patterns. For more details on $\mathcal{L}_{\mathrm{td}}$ loss, we refer the readers to (Xiong et al., 2024).

In our experiments, we set $\mathcal{L}_{\mathrm{err}}$ to Mean Absolute Error (MAE) for the benchmarking tasks. We hypothesize that MAE is more suitable because *DeepEDM* aims to model the underlying dynamics and thus focusing on the general trend aligns better with this objective, avoiding excessive sensitivity to noisy outliers. However, for ECL and Traffic datasets which have high dimensionality and are generally noisier, MAE does not perform competitively. For these specific cases, we instead set $\mathcal{L}_{\mathrm{err}}$ to Mean Squared Error (MSE) loss.

**Training:** *DeepEDM* is trained for 250 epochs using the AdamW (Loshchilov & Hutter, 2017) optimizer with a learning rate of 0.0005 and a batch size of 32. Following standard practices in time-series forecasting, an early stopping mechanism based on validation set performance metrics is implemented to mitigate overfitting.

### A.3. Short-term Forecasting Experiments and Results

We now present the comprehensive evaluation results of the *DeepEDM* on the popular short-term univariate forecasting M4 benchmark. This benchmark consists of six datasets, each corresponding to a different frequency: *yearly*, *quarterly*, *monthly*, *weekly*, *daily*, and *hourly*. For our experiments, we follow the standard setup of all the reported baselines, where the lookback length is set to twice the forecast length $H \in [6, 48]$. Consistent with prior works, *DeepEDM* is optimized using the *SMAPE* loss function.

**Metrics:** Following standard baselines, we use the Symmetric Mean Absolute Percentage Error (SMAPE), MAPE (Mean Absolute Percentage Error), Mean Absolute Scaled Error (MASE), and overall weighted average (OWA) metrics to evaluate the forecasting performance. For brevity, we only provide the formulation of these metrics and refer the reader to (Oreshkin et al., 2020) for more details:

$$\mathrm{SMAPE} = \frac{200}{H} \sum_{i=1}^{H} \frac{|y_{T+i} - \hat{y}_{T+i}|}{|y_{T+i}| + |\hat{y}_{T+i}|}, \quad \mathrm{MAPE} = \frac{100}{H} \sum_{i=1}^{H} \frac{|y_{T+i} - \hat{y}_{T+i}|}{|y_{T+i}|},$$

$$\mathrm{MASE} = \frac{1}{H} \sum_{i=1}^{H} \frac{|y_{T+i} - \hat{y}_{T+i}|}{\frac{1}{T+H-m} \sum_{j=m+1}^{T+H} |y_j - y_{j-m}|}, \quad \mathrm{OWA} = \frac{1}{2} \left[ \frac{\mathrm{SMAPE}}{\mathrm{SMAPE}_{\mathrm{Naïve2}}} + \frac{\mathrm{MASE}}{\mathrm{MASE}_{\mathrm{Naïve2}}} \right].$$

**Results:** The results of our experiments, summarized in Table 5, demonstrate that *DeepEDM* outperforms the dynamical modeling-based methods on all subsets. It also surpasses all methods within the three subsets grouped under the "others"

category, while delivering competitive performance across other subsets. The notable success in the "others" category can be attributed to the typically longer sequences found in these subsets, which better facilitate the reconstruction of the underlying dynamical system. In contrast, other subsets contain shorter sequences, which pose challenges to effective system reconstruction. Nonetheless, *DeepEDM* again exhibits competitive performance (best Weighted Average SMAPE for all datasets) across this benchmark, further showcasing its capabilities.

*Table 5.* Univariate forecasting results on M4 dataset. The M4 dataset comprises six datasets, three of which are included in the "Others" category. These three subsets generally contain longer sequences, allowing our method to perform better and achieve superior performance compared to all other methods on these subsets. All prediction lengths are in $[6, 48]$. Baseline results are from Koopa (2024b) and TimeMixer (2024a). **Bold** represents the best values while underline represents 2nd best. Gray represents dynamical modeling based methods.

| | Models | **DeepEDM** | Koopa | KNF | TimeMixer | TimesNet | N-HiTS | N-BEATS* | PatchTST | FiLM | LightTS | DLinear | FED. | Stationary | Auto. | Pyra. | In. |
|---|---|---|---|---|---|---|---|---|---|---|---|---|---|---|---|---|---|
| **Yearly** | SMAPE | 13.243 | 13.352 | 13.986 | **13.206** | 13.387 | 13.418 | 13.436 | 16.463 | 17.431 | 14.247 | 16.965 | 13.728 | 13.717 | 13.974 | 15.530 | 14.727 |
| | MASE | 2.973 | 2.997 | 3.029 | **2.916** | 2.996 | 3.045 | 3.043 | 3.967 | 4.043 | 3.109 | 4.283 | 3.048 | 3.078 | 3.134 | 3.711 | 3.418 |
| | OWA | 0.779 | 0.786 | 0.804 | **0.776** | 0.786 | 0.793 | 0.794 | 1.003 | 1.042 | 0.827 | 1.058 | 0.803 | 0.807 | 0.822 | 0.942 | 0.881 |
| **Quarterly** | SMAPE | 10.04 | 10.159 | 10.343 | **9.996** | 10.100 | 10.202 | 10.124 | 10.644 | 12.925 | 11.364 | 12.145 | 10.792 | 10.958 | 11.338 | 15.449 | 11.360 |
| | MASE | 1.177 | 1.189 | 1.202 | **1.166** | 1.182 | 1.194 | 1.169 | 1.278 | 1.664 | 1.328 | 1.520 | 1.283 | 1.325 | 1.365 | 2.350 | 1.401 |
| | OWA | 0.885 | 0.895 | 0.965 | **0.825** | 0.890 | 0.899 | 0.886 | 0.949 | 1.193 | 1.000 | 1.106 | 0.958 | 0.981 | 1.012 | 1.558 | 1.027 |
| **Monthly** | SMAPE | **12.547** | 12.730 | 12.894 | 12.605 | 12.670 | 12.791 | 12.677 | 13.399 | 15.407 | 14.014 | 13.514 | 14.260 | 13.917 | 13.958 | 17.642 | 14.062 |
| | MASE | 0.933 | 0.953 | 1.023 | **0.919** | 0.933 | 0.969 | 0.937 | 1.031 | 1.298 | 1.053 | 1.037 | 1.102 | 1.097 | 1.103 | 1.913 | 1.141 |
| | OWA | 0.873 | 0.901 | 0.985 | **0.869** | 0.878 | 0.899 | 0.880 | 0.949 | 1.144 | 0.981 | 0.956 | 1.012 | 0.998 | 1.002 | 1.511 | 1.024 |
| **Others** | SMAPE | **4.339** | 4.861 | 4.753 | 4.564 | 4.891 | 5.061 | 4.925 | 6.558 | 7.134 | 15.880 | 6.709 | 4.954 | 6.302 | 5.485 | 24.786 | 24.460 |
| | MASE | 3.042 | 3.124 | 3.138 | 3.115 | 3.302 | 3.216 | 3.391 | 4.511 | 5.09 | 11.434 | 4.953 | 3.264 | 4.064 | 3.865 | 18.581 | 20.960 |
| | OWA | **0.936** | 1.004 | 1.019 | 0.982 | 1.035 | 1.040 | 1.053 | 1.401 | 1.553 | 3.474 | 1.487 | 1.036 | 1.304 | 1.187 | 5.538 | 5.879 |
| **Weighted Average** | SMAPE | **11.695** | 11.863 | 12.126 | 11.723 | 11.829 | 11.927 | 11.851 | 13.152 | 14.863 | 13.525 | 13.639 | 12.840 | 12.780 | 12.909 | 16.987 | 14.086 |
| | MASE | 1.566 | 1.595 | 1.641 | **1.559** | 1.585 | 1.613 | 1.559 | 1.945 | 2.207 | 2.111 | 2.095 | 1.701 | 1.756 | 1.771 | 3.265 | 2.718 |
| | OWA | 0.841 | 0.858 | 0.874 | **0.840** | 0.851 | 0.861 | 0.855 | 0.998 | 1.125 | 1.051 | 1.051 | 0.918 | 0.930 | 0.939 | 1.480 | 1.230 |

∗ The original paper of N-BEATS (2020) adopts a special ensemble method to promote the performance. For fair comparison, authors of TimeMixer (2024a) removed the ensemble and only compared the pure forecasting models.

## A.4. Long-term Forecasting with Lookback Search

In this section, we present the forecasting results under the lookback search setting, commonly adopted in recent works such as *TimeMixer* (Wang et al., 2024a). This setting allows models to select an optimal lookback length from a predefined set, ensuring a fair comparison while potentially benefiting methods that can leverage longer historical dependencies. In this setting, each model is evaluated on four prediction horizons ($H \in [96, 192, 336, 720]$), with the best-performing lookback chosen from $[96, 192, 336, 512]$.

While this setup provides flexibility, it can be particularly challenging for dynamical systems-based methods like DeepEDM, which rely on sufficiently long lookbacks to reconstruct the underlying attractor accurately. Some configurations require forecasting 720 steps into the future using only 512 steps of history, a scenario that may not always capture the full state-space dynamics. Nonetheless, as shown in Table 6, *DeepEDM* demonstrates strong performance, achieving *45* wins compared to 34 for the second-best model, further highlighting its robustness even under challenging settings.

## A.5. Ablation Studies

In addition to the component-wise ablation presented in the main paper, we conduct further experiments to evaluate additional design choices underlying DeepEDM.

### A.5.1. ABLATION STUDY ON LOOKBACK LENGTH

In this section, we present an ablation study to investigate the impact of varying the lookback length $T$ on forecasting performance, evaluated across three datasets: *ETTh1*, *ETTm2*, and *Exchange*. The results, shown in Figure 4 plot the Mean Squared Error (MSE) against input sequence lengths, with the prediction horizon fixed at $H = 96$. Across the datasets, we observe that increasing the lookback window improves forecasting accuracy up to a threshold, typically at $T = 512$. Beyond this point, performance degrades, with further increases in lookback length resulting in higher errors. This decline is attributed to a *distribution shift*, where the model starts to capture data points from the past that no longer align with the distribution of more recent data, introducing *irrelevant or outdated information* that negatively impacts forecast quality.

Notably, our model consistently outperforms the compared benchmarks across varying input sequence lengths, demonstrating

*Table 6.* Multivariate forecasting results under the lookback search setting. **Bold** indicates the best performance, while underline indicates the 2nd best. Baseline results are taken from (Wang et al., 2024a) while Naïve was reproduced by us.

| Models | | Ours MSE | Ours MAE | TimeMixer MSE | TimeMixer MAE | PatchTST MSE | PatchTST MAE | TimesNet MSE | TimesNet MAE | Crossformer MSE | Crossformer MAE | MICN MSE | MICN MAE | FiLM MSE | FiLM MAE | DLinear MSE | DLinear MAE | FEDformer MSE | FEDformer MAE | Stationary MSE | Stationary MAE | Autoformer MSE | Autoformer MAE | Naïve MSE | Naïve MAE |
|---|---|---|---|---|---|---|---|---|---|---|---|---|---|---|---|---|---|---|---|---|---|---|---|---|---|
| Weather | 96 | **0.145** | **0.183** | 0.147 | 0.197 | 0.149 | 0.198 | 0.172 | 0.220 | 0.232 | 0.302 | 0.161 | 0.229 | 0.199 | 0.262 | 0.176 | 0.237 | 0.217 | 0.296 | 0.173 | 0.223 | 0.266 | 0.336 | 0.259 | 0.254 |
| | 192 | **0.189** | **0.226** | 0.189 | 0.239 | 0.194 | 0.241 | 0.219 | 0.261 | 0.371 | 0.410 | 0.220 | 0.281 | 0.228 | 0.288 | 0.220 | 0.282 | 0.276 | 0.336 | 0.245 | 0.285 | 0.307 | 0.367 | 0.309 | 0.292 |
| | 336 | **0.240** | **0.267** | 0.241 | 0.280 | 0.306 | 0.282 | 0.246 | 0.337 | 0.495 | 0.515 | 0.278 | 0.331 | 0.267 | 0.323 | 0.265 | 0.319 | 0.339 | 0.380 | 0.321 | 0.338 | 0.359 | 0.395 | 0.376 | 0.338 |
| | 720 | 0.314 | **0.322** | 0.310 | 0.330 | 0.314 | 0.334 | 0.365 | 0.359 | 0.526 | 0.542 | 0.311 | 0.356 | 0.319 | 0.361 | 0.323 | 0.362 | 0.403 | 0.428 | 0.414 | 0.410 | 0.419 | 0.428 | 0.465 | 0.394 |
| | Avg | **0.222** | **0.249** | 0.222 | 0.262 | 0.241 | 0.264 | 0.251 | 0.294 | 0.406 | 0.442 | 0.242 | 0.299 | 0.253 | 0.309 | 0.246 | 0.300 | 0.309 | 0.360 | 0.288 | 0.314 | 0.338 | 0.382 | 0.352 | 0.319 |
| Solar-Energy | 96 | 0.178 | **0.199** | 0.167 | 0.220 | 0.224 | 0.278 | 0.219 | 0.314 | 0.181 | 0.240 | 0.188 | 0.252 | 0.320 | 0.339 | 0.289 | 0.377 | 0.201 | 0.304 | 0.321 | 0.380 | 0.456 | 0.446 | 1.539 | 0.816 |
| | 192 | 0.191 | 0.209 | 0.187 | 0.249 | 0.253 | 0.298 | 0.231 | 0.322 | 0.196 | 0.252 | 0.215 | 0.280 | 0.360 | 0.362 | 0.319 | 0.397 | 0.237 | 0.337 | 0.346 | 0.369 | 0.588 | 0.561 | 1.360 | 0.735 |
| | 336 | 0.206 | 0.216 | 0.200 | 0.258 | 0.273 | 0.306 | 0.246 | 0.337 | 0.216 | 0.243 | 0.222 | 0.267 | 0.398 | 0.375 | 0.352 | 0.415 | 0.254 | 0.362 | 0.357 | 0.387 | 0.595 | 0.588 | 1.430 | 0.766 |
| | 720 | 0.254 | 0.245 | 0.215 | 0.250 | 0.272 | 0.308 | 0.280 | 0.363 | 0.220 | 0.256 | 0.226 | 0.264 | 0.399 | 0.368 | 0.356 | 0.412 | 0.280 | 0.397 | 0.335 | 0.384 | 0.733 | 0.633 | 1.474 | 0.784 |
| | Avg | 0.207 | **0.217** | 0.192 | 0.244 | 0.256 | 0.298 | 0.244 | 0.334 | 0.204 | 0.248 | 0.213 | 0.266 | 0.369 | 0.361 | 0.329 | 0.400 | 0.243 | 0.350 | 0.340 | 0.380 | 0.593 | 0.557 | 1.451 | 0.775 |
| Electricity | 96 | 0.133 | 0.228 | 0.129 | 0.224 | 0.129 | 0.222 | 0.168 | 0.272 | 0.150 | 0.251 | 0.164 | 0.269 | 0.154 | 0.267 | 0.140 | 0.237 | 0.193 | 0.308 | 0.169 | 0.273 | 0.201 | 0.317 | 1.588 | 0.946 |
| | 192 | 0.151 | 0.246 | 0.140 | 0.220 | 0.147 | 0.240 | 0.184 | 0.322 | 0.161 | 0.260 | 0.177 | 0.285 | 0.164 | 0.258 | 0.153 | 0.249 | 0.201 | 0.315 | 0.182 | 0.286 | 0.222 | 0.334 | 1.596 | 0.951 |
| | 336 | 0.167 | 0.262 | 0.161 | 0.255 | 0.163 | 0.259 | 0.198 | 0.300 | 0.182 | 0.281 | 0.193 | 0.304 | 0.188 | 0.283 | 0.169 | 0.267 | 0.214 | 0.329 | 0.200 | 0.304 | 0.231 | 0.338 | 1.618 | 0.961 |
| | 720 | 0.205 | 0.293 | 0.194 | 0.287 | 0.197 | 0.290 | 0.220 | 0.320 | 0.251 | 0.339 | 0.212 | 0.321 | 0.236 | 0.332 | 0.203 | 0.301 | 0.246 | 0.355 | 0.222 | 0.321 | 0.254 | 0.361 | 1.647 | 0.975 |
| | Avg | 0.164 | 0.257 | 0.156 | 0.246 | 0.159 | 0.253 | 0.192 | 0.295 | 0.186 | 0.283 | 0.186 | 0.295 | 0.186 | 0.285 | 0.166 | 0.264 | 0.214 | 0.321 | 0.213 | 0.296 | 0.227 | 0.338 | 1.612 | 0.958 |
| Traffic | 96 | **0.360** | 0.252 | 0.360 | 0.249 | 0.360 | 0.249 | 0.593 | 0.321 | 0.514 | 0.267 | 0.519 | 0.309 | 0.416 | 0.294 | 0.410 | 0.282 | 0.587 | 0.366 | 0.612 | 0.338 | 0.613 | 0.388 | 2.715 | 1.077 |
| | 192 | **0.375** | 0.255 | 0.375 | 0.250 | 0.379 | 0.256 | 0.617 | 0.336 | 0.549 | 0.252 | 0.537 | 0.315 | 0.408 | 0.288 | 0.423 | 0.287 | 0.604 | 0.373 | 0.613 | 0.340 | 0.616 | 0.382 | 2.747 | 1.085 |
| | 336 | 0.410 | 0.280 | 0.385 | 0.270 | 0.392 | 0.264 | 0.629 | 0.336 | 0.530 | 0.300 | 0.534 | 0.313 | 0.425 | 0.298 | 0.436 | 0.296 | 0.621 | 0.383 | 0.618 | 0.328 | 0.622 | 0.337 | 2.788 | 1.094 |
| | 720 | 0.458 | 0.311 | 0.430 | 0.281 | 0.432 | 0.286 | 0.640 | 0.350 | 0.573 | 0.313 | 0.577 | 0.325 | 0.520 | 0.353 | 0.466 | 0.315 | 0.626 | 0.382 | 0.653 | 0.355 | 0.660 | 0.408 | 2.810 | 1.097 |
| | Avg | 0.401 | 0.275 | 0.387 | 0.262 | 0.391 | 0.264 | 0.620 | 0.336 | 0.542 | 0.283 | 0.541 | 0.315 | 0.442 | 0.308 | 0.434 | 0.295 | 0.609 | 0.376 | 0.624 | 0.340 | 0.628 | 0.379 | 2.765 | 1.088 |
| ETTh1 | 96 | **0.356** | **0.384** | 0.361 | 0.390 | 0.370 | 0.400 | 0.384 | 0.402 | 0.418 | 0.438 | 0.421 | 0.431 | 0.422 | 0.432 | 0.375 | 0.399 | 0.376 | 0.419 | 0.513 | 0.491 | 0.449 | 0.459 | 1.294 | 0.713 |
| | 192 | **0.398** | 0.417 | 0.409 | 0.414 | 0.413 | 0.429 | 0.436 | 0.429 | 0.539 | 0.517 | 0.474 | 0.487 | 0.462 | 0.458 | 0.405 | 0.416 | 0.420 | 0.448 | 0.534 | 0.504 | 0.500 | 0.482 | 1.325 | 0.733 |
| | 336 | **0.419** | **0.425** | 0.430 | 0.429 | 0.422 | 0.440 | 0.638 | 0.469 | 0.709 | 0.638 | 0.569 | 0.551 | 0.501 | 0.483 | 0.439 | 0.443 | 0.459 | 0.465 | 0.588 | 0.535 | 0.521 | 0.496 | 1.330 | 0.746 |
| | 720 | **0.434** | **0.451** | 0.445 | 0.460 | 0.447 | 0.468 | 0.521 | 0.500 | 0.733 | 0.636 | 0.770 | 0.672 | 0.544 | 0.526 | 0.472 | 0.490 | 0.506 | 0.507 | 0.643 | 0.616 | 0.514 | 0.512 | 1.335 | 0.755 |
| | Avg | **0.402** | **0.419** | 0.411 | 0.423 | 0.413 | 0.434 | 0.458 | 0.450 | 0.600 | 0.557 | 0.558 | 0.535 | 0.482 | 0.475 | 0.423 | 0.437 | 0.440 | 0.460 | 0.570 | 0.536 | 0.496 | 0.487 | 1.321 | 0.737 |
| ETTh2 | 96 | 0.275 | 0.332 | 0.271 | 0.330 | 0.274 | 0.337 | 0.340 | 0.374 | 0.425 | 0.463 | 0.299 | 0.364 | 0.323 | 0.370 | 0.289 | 0.353 | 0.346 | 0.388 | 0.476 | 0.458 | 0.358 | 0.397 | 0.432 | 0.422 |
| | 192 | 0.341 | 0.374 | 0.317 | 0.402 | 0.314 | 0.382 | 0.231 | 0.322 | 0.473 | 0.500 | 0.441 | 0.454 | 0.391 | 0.415 | 0.383 | 0.418 | 0.429 | 0.439 | 0.512 | 0.493 | 0.456 | 0.452 | 0.534 | 0.472 |
| | 336 | 0.360 | 0.393 | 0.332 | 0.396 | 0.329 | 0.384 | 0.452 | 0.452 | 0.581 | 0.562 | 0.654 | 0.567 | 0.415 | 0.440 | 0.448 | 0.465 | 0.496 | 0.487 | 0.552 | 0.551 | 0.482 | 0.486 | 0.597 | 0.511 |
| | 720 | 0.386 | 0.424 | 0.342 | 0.408 | 0.379 | 0.422 | 0.462 | 0.468 | 0.775 | 0.665 | 0.956 | 0.716 | 0.441 | 0.459 | 0.605 | 0.551 | 0.463 | 0.474 | 0.562 | 0.560 | 0.515 | 0.511 | 0.595 | 0.519 |
| | Avg | 0.341 | 0.381 | 0.316 | 0.384 | 0.324 | 0.381 | 0.371 | 0.404 | 0.564 | 0.548 | 0.588 | 0.525 | 0.393 | 0.421 | 0.431 | 0.447 | 0.433 | 0.447 | 0.526 | 0.516 | 0.453 | 0.462 | 0.539 | 0.481 |
| ETTm1 | 96 | **0.289** | **0.331** | 0.291 | 0.340 | 0.293 | 0.346 | 0.338 | 0.375 | 0.361 | 0.403 | 0.316 | 0.362 | 0.302 | 0.345 | 0.299 | 0.343 | 0.379 | 0.419 | 0.386 | 0.398 | 0.505 | 0.475 | 1.214 | 0.665 |
| | 192 | **0.321** | **0.351** | 0.327 | 0.365 | 0.333 | 0.370 | 0.374 | 0.387 | 0.387 | 0.422 | 0.363 | 0.390 | 0.338 | 0.368 | 0.335 | 0.365 | 0.426 | 0.441 | 0.459 | 0.444 | 0.553 | 0.496 | 1.261 | 0.690 |
| | 336 | 0.361 | **0.377** | 0.360 | 0.381 | 0.369 | 0.392 | 0.410 | 0.411 | 0.605 | 0.572 | 0.408 | 0.426 | 0.373 | 0.388 | 0.369 | 0.386 | 0.445 | 0.459 | 0.495 | 0.464 | 0.621 | 0.537 | 1.287 | 0.707 |
| | 720 | **0.414** | **0.406** | 0.415 | 0.417 | 0.416 | 0.420 | 0.478 | 0.450 | 0.703 | 0.645 | 0.481 | 0.476 | 0.420 | 0.420 | 0.425 | 0.421 | 0.543 | 0.490 | 0.585 | 0.516 | 0.671 | 0.561 | 1.322 | 0.730 |
| | Avg | **0.346** | **0.366** | 0.348 | 0.375 | 0.353 | 0.382 | 0.353 | 0.382 | 0.514 | 0.510 | 0.392 | 0.413 | 0.358 | 0.380 | 0.357 | 0.379 | 0.448 | 0.452 | 0.481 | 0.456 | 0.588 | 0.517 | 1.271 | 0.698 |
| ETTm2 | 96 | **0.164** | **0.245** | 0.164 | 0.254 | 0.166 | 0.256 | 0.187 | 0.267 | 0.275 | 0.358 | 0.179 | 0.275 | 0.165 | 0.256 | 0.167 | 0.260 | 0.203 | 0.287 | 0.192 | 0.274 | 0.255 | 0.339 | 0.267 | 0.328 |
| | 192 | **0.221** | **0.287** | 0.223 | 0.295 | 0.223 | 0.296 | 0.249 | 0.309 | 0.345 | 0.400 | 0.307 | 0.376 | 0.222 | 0.296 | 0.224 | 0.303 | 0.269 | 0.328 | 0.280 | 0.339 | 0.281 | 0.340 | 0.340 | 0.371 |
| | 336 | **0.270** | **0.321** | 0.279 | 0.330 | 0.274 | 0.329 | 0.321 | 0.351 | 0.657 | 0.528 | 0.325 | 0.388 | 0.277 | 0.333 | 0.281 | 0.342 | 0.325 | 0.366 | 0.334 | 0.361 | 0.339 | 0.372 | 0.412 | 0.410 |
| | 720 | **0.347** | **0.371** | 0.359 | 0.383 | 0.362 | 0.385 | 0.408 | 0.403 | 1.208 | 0.753 | 0.502 | 0.490 | 0.371 | 0.389 | 0.397 | 0.421 | 0.421 | 0.415 | 0.417 | 0.413 | 0.422 | 0.419 | 0.522 | 0.466 |
| | Avg | **0.251** | **0.306** | 0.256 | 0.315 | 0.256 | 0.317 | 0.291 | 0.333 | 0.621 | 0.510 | 0.328 | 0.382 | 0.259 | 0.319 | 0.267 | 0.332 | 0.304 | 0.349 | 0.306 | 0.347 | 0.324 | 0.368 | 0.385 | 0.393 |
| 1st Count | | 45 | | 34 | | 8 | | 2 | | 0 | | 0 | | 0 | | 0 | | 0 | | 0 | | 0 | | 0 | |

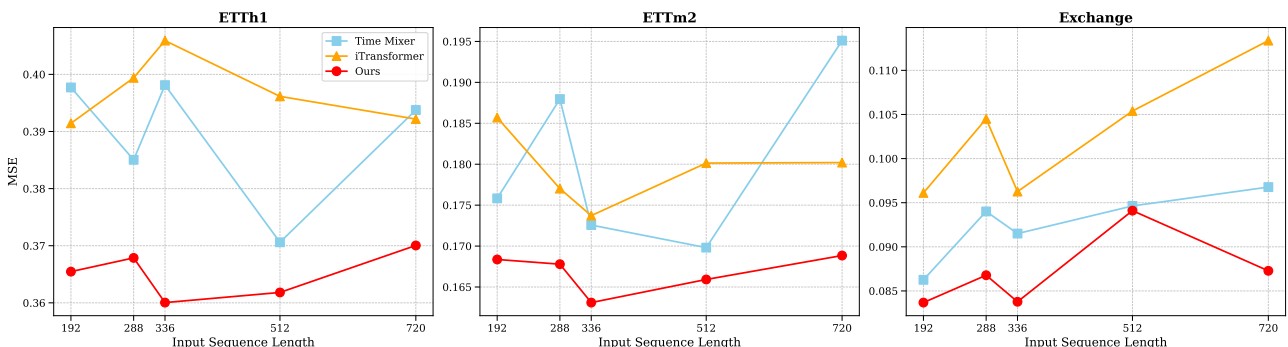

*Figure 4.* Impact of lookback length on forecast accuracy. The prediction horizon is fixed at $H = 96$, while the input sequence length $T \in \{192, 288, 336, 512, 720\}$ is varied to assess its effect on forecasting performance. Increasing the lookback window generally improves accuracy up to a certain point; however, excessively long lookbacks can introduce irrelevant information or noise, ultimately degrading performance.

its robustness in different temporal contexts. This study highlights the importance of a *balanced lookback window* in optimizing forecasting models. While short windows may lack sufficient context, excessively long windows risk overfitting to irrelevant historical trends. Our results suggest that a moderate lookback window length (e.g., $T = 512$) offers the best trade-off between context and relevance, as evidenced by the performance drop beyond this threshold.

### A.5.2. SENSITIVITY TO TIME DELAY AND EMBEDDING DIMENSION

We conducted additional experiments to investigate the sensitivity of *DeepEDM* to two key hyperparameters that govern the time-delay embedding: the embedding dimension $\delta_T$ and the delay interval $\tau$. Together, these parameters define how historical observations are mapped into the delay-coordinate space. Specifically, a time-delay embedding with $\delta_T = 3$ and $\tau = 1$ yields a 3-dimensional vector $\mathbf{x}_t = [x_t, x_{t-1}, x_{t-2}]$, while $\delta_T = 3$ and $\tau = 2$ results in $\mathbf{x}_t = [x_t, x_{t-2}, x_{t-4}]$.

In our main experiments, $\tau$ is fixed to 1, and $\delta_T$ is selected empirically per dataset. To assess the robustness of the model, we perform ablation studies by varying one parameter while keeping the other fixed. For the $\delta_T$-sensitivity experiments, we fix $\tau = 1$ and vary $\delta_T \in \{2, 3, 4, 5\}$. For the $\tau$-sensitivity experiments, we fix $\delta_T = 3$ and explore $\tau \in \{1, 2, 3, 4\}$.

**Effect of Embedding Dimension $\delta_T$.** The impact of the embedding dimension $\delta_T$ is summarized in Table 7. The results show that its influence is highly dataset-dependent. In several cases, performance remains relatively stable across different values of $\delta_T$, indicating that the underlying system may be either intrinsically low-dimensional or already adequately represented by the chosen embedding. When the dynamics lie on a low-dimensional manifold, larger values of $\delta_T$ become redundant. Conversely, in high-dimensional systems, small $\delta_T$ may lead to underembedding, resulting in similar but suboptimal performance across configurations. These observations align with classical results from delay-coordinate embedding theory.

**Effect of Delay Interval $\tau$.** Table 8 presents the results for varying the delay interval $\tau$. We observe that $\tau = 1$ consistently yields the best or near-best performance across all datasets. This is also the case considered in Takens' theorem. Nevertheless, determining the optimal pair $(\delta_T, \tau)$ remains an open problem. Developing principled strategies for joint selection may further improve model accuracy, which we leave for future work.

### A.5.3. ABLATION STUDY ON THE LOSS FUNCTION

To study the role of the loss function, we compare the full *DeepEDM* model trained with our full loss function to a variant trained solely with standard MSE. The results of this experiment, detailed in Table 9, reveal that while incoporating $\mathcal{L}_{td}$ loss generally leads to lower errors, MSE occasionally performs comparably or even slightly better—suggesting complementary strengths. Importantly, the full model consistently delivers the best overall performance.

*Table 7.* Ablation on $m$ i.e. embedding size

| Dataset | | $\delta_T = 1$ | | $\delta_T = 5$ | | $\delta_T = 7$ | | $\delta_T = 11$ | | $\delta_T = 15$ | |
|---|---|---|---|---|---|---|---|---|---|---|---|
| | | MSE | MAE | MSE | MAE | MSE | MAE | MSE | MAE | MSE | MAE |
| ETTh1 | 48 | 0.3288 | 0.3587 | **0.3224** | **0.3561** | 0.3245 | 0.3576 | 0.3240 | 0.3582 | 0.3236 | 0.3583 |
| | 96 | **0.3656** | **0.3835** | 0.3708 | 0.3865 | 0.3688 | 0.3843 | 0.3686 | 0.3858 | 0.3715 | 0.3866 |
| | 144 | 0.3922 | 0.3993 | **0.3881** | **0.3978** | 0.3886 | 0.3980 | 0.3915 | 0.3995 | 0.3918 | 0.4000 |
| | 192 | 0.4064 | 0.4152 | 0.4046 | 0.4147 | **0.4017** | **0.4132** | 0.4258 | 0.4327 | 0.4593 | 0.4526 |
| ETTh2 | 48 | 0.2263 | 0.2886 | 0.2276 | 0.2902 | 0.2238 | 0.2871 | **0.2231** | 0.2872 | 0.2236 | **0.2869** |
| | 96 | 0.2917 | 0.3341 | **0.2870** | **0.3320** | 0.2881 | 0.3328 | 0.2905 | 0.3335 | 0.2940 | 0.3344 |
| | 144 | 0.3307 | 0.3635 | 0.3225 | **0.3603** | **0.3224** | 0.3611 | 0.3270 | 0.3631 | 0.3352 | 0.3666 |
| | 192 | 0.3523 | 0.3783 | 0.3516 | 0.3764 | **0.3450** | **0.3741** | 0.3505 | 0.3763 | 0.3451 | 0.3765 |
| ETTm1 | 48 | 0.2822 | 0.3223 | 0.2827 | 0.3224 | **0.2775** | 0.3191 | 0.2782 | **0.3175** | 0.2784 | 0.3182 |
| | 96 | 0.2902 | 0.3287 | **0.2874** | **0.3273** | 0.2878 | 0.3277 | 0.2904 | 0.3290 | 0.2904 | 0.3280 |
| | 144 | 0.3104 | 0.3441 | 0.3084 | 0.3445 | 0.3041 | 0.3428 | 0.3087 | 0.3439 | **0.3034** | **0.3422** |
| | 192 | 0.3226 | 0.3533 | **0.3200** | **0.3523** | 0.3213 | 0.3524 | 0.3257 | 0.3545 | 0.3226 | 0.3537 |
| ETTm2 | 48 | 0.1344 | 0.2219 | 0.1334 | 0.2213 | 0.1335 | **0.2211** | **0.1332** | 0.2212 | 0.1337 | **0.2211** |
| | 96 | 0.1689 | 0.2473 | 0.1692 | 0.2478 | 0.1697 | 0.2483 | 0.1691 | 0.2478 | **0.1670** | **0.2470** |
| | 144 | 0.1991 | 0.2707 | 0.2048 | 0.2727 | **0.1974** | **0.2685** | 0.2023 | 0.2722 | 0.2057 | 0.2727 |
| | 192 | 0.2272 | 0.2886 | **0.2244** | 0.2890 | 0.2248 | **0.2881** | 0.2288 | 0.2904 | 0.2291 | 0.2911 |
| Exchange | 48 | 0.0429 | 0.1429 | **0.0418** | **0.1404** | 0.0429 | 0.1414 | 0.0443 | 0.1458 | 0.0430 | 0.1433 |
| | 96 | 0.0894 | 0.2087 | 0.0854 | 0.2026 | **0.0825** | **0.2008** | 0.0861 | 0.2039 | 0.0886 | 0.2061 |
| | 144 | 0.1348 | 0.2586 | **0.1298** | **0.2529** | 0.1326 | 0.2553 | 0.1427 | 0.2664 | 0.1323 | 0.2560 |
| | 192 | 0.1801 | 0.3035 | 0.1931 | 0.3144 | 0.1785 | **0.2996** | 0.1854 | 0.3085 | **0.1777** | 0.3000 |
| Weather | 48 | 0.1404 | 0.1695 | 0.1421 | 0.1735 | **0.1371** | **0.1668** | 0.1396 | 0.1686 | 0.1424 | 0.1753 |
| | 96 | 0.1600 | 0.1948 | **0.1573** | **0.1916** | 0.1579 | 0.1927 | 0.1578 | 0.1920 | 0.1578 | 0.1919 |
| | 144 | 0.1741 | 0.2099 | **0.1735** | **0.2092** | 0.1741 | 0.2096 | 0.1749 | 0.2098 | 0.1744 | 0.2094 |
| | 192 | **0.1910** | **0.2262** | 0.1910 | 0.2263 | 0.1911 | 0.2263 | 0.1911 | 0.2263 | 0.1911 | 0.2264 |

*Table 8.* Ablation on $\tau$.

| Dataset | | $\tau = 1$ | | $\tau = 2$ | | $\tau = 3$ | |
|---|---|---|---|---|---|---|---|
| | | MSE | MAE | MSE | MAE | MSE | MAE |
| ETTh1 | 48 | **0.3246** | **0.3575** | 0.3270 | 0.3589 | 0.3270 | 0.3602 |
| | 96 | 0.3689 | 0.3844 | 0.3663 | 0.3839 | **0.3653** | **0.3836** |
| | 144 | **0.3885** | **0.3980** | 0.3902 | 0.3988 | 0.3911 | 0.3985 |
| | 192 | **0.4017** | **0.4129** | 0.4038 | 0.4164 | 0.4090 | 0.4194 |
| ETTh2 | 48 | **0.2238** | 0.2871 | 0.2269 | 0.2879 | 0.2240 | **0.2870** |
| | 96 | 0.2882 | 0.3333 | 0.2885 | 0.3331 | **0.2875** | **0.3326** |
| | 144 | **0.3224** | **0.3611** | 0.3290 | 0.3652 | 0.3316 | 0.3653 |
| | 192 | **0.3450** | **0.3741** | 0.3541 | 0.3787 | 0.3583 | 0.3803 |
| ETTm1 | 48 | **0.2782** | 0.3195 | 0.2795 | **0.3189** | 0.2804 | 0.3198 |
| | 96 | 0.2869 | 0.3282 | 0.2901 | 0.3288 | **0.2866** | **0.3279** |
| | 144 | **0.3053** | **0.3439** | 0.3094 | 0.3457 | 0.3079 | 0.3452 |
| | 192 | 0.3213 | **0.3523** | 0.3209 | 0.3527 | **0.3203** | 0.3530 |
| ETTm2 | 48 | **0.1337** | **0.2212** | 0.1342 | 0.2222 | 0.1346 | 0.2230 |
| | 96 | **0.1700** | 0.2485 | 0.1704 | 0.2485 | 0.1706 | **0.2479** |
| | 144 | 0.1965 | 0.2684 | 0.1967 | 0.2690 | **0.1952** | **0.2681** |
| | 192 | **0.2220** | **0.2871** | 0.2246 | 0.2881 | 0.2231 | 0.2873 |
| Exchange | 48 | 0.0429 | 0.1414 | 0.0426 | 0.1412 | **0.0423** | **0.1410** |
| | 96 | **0.0829** | **0.2011** | 0.0835 | 0.2020 | 0.0838 | 0.2023 |
| | 144 | 0.1340 | 0.2567 | 0.1336 | 0.2564 | **0.1317** | **0.2547** |
| | 192 | **0.1769** | **0.2991** | 0.1773 | 0.2992 | 0.1789 | 0.2999 |
| Weather | 48 | **0.1371** | **0.1668** | 0.1397 | 0.1690 | 0.1406 | 0.1695 |
| | 96 | 0.1581 | **0.1926** | **0.1580** | 0.1931 | 0.1587 | 0.1938 |
| | 144 | 0.1741 | **0.2096** | **0.1740** | **0.2096** | 0.1749 | 0.2110 |
| | 192 | **0.1938** | **0.2280** | 0.1969 | 0.2310 | 0.1956 | 0.2308 |

*Table 9.* Ablation study on the loss function.

| Dataset | | DeepEDM | | DeepEDM (with MSE loss) | |
|---|---|---|---|---|---|
| | | MSE | MAE | MSE | MAE |
| ECL | 48 | 0.1610 | 0.2470 | **0.1591** | **0.2463** |
| | 96 | **0.1370** | **0.2320** | 0.1380 | 0.2322 |
| | 144 | **0.1450** | **0.2390** | 0.1466 | 0.2393 |
| | 192 | **0.1510** | **0.2440** | 0.1528 | 0.2458 |
| ETTh1 | 48 | **0.3240** | **0.3570** | 0.3322 | 0.3738 |
| | 96 | **0.3650** | **0.3840** | 0.3697 | 0.3982 |
| | 144 | **0.3880** | **0.3980** | 0.3990 | 0.4095 |
| | 192 | 0.4070 | 0.4210 | **0.4061** | **0.4205** |
| Exchange | 48 | 0.0420 | 0.1420 | **0.0415** | **0.1403** |
| | 96 | 0.0880 | 0.2050 | **0.0827** | **0.2025** |
| | 144 | 0.1330 | **0.2550** | **0.1297** | 0.2552 |
| | 192 | 0.1780 | 0.3010 | **0.1739** | **0.3000** |
| ETTh2 | 48 | **0.2250** | **0.2880** | 0.2265 | 0.2958 |
| | 96 | 0.2890 | **0.3330** | **0.2885** | 0.3430 |
| | 144 | **0.3240** | **0.3620** | 0.3242 | 0.3679 |
| | 192 | **0.3510** | **0.3770** | 0.3547 | 0.3888 |
| Traffic | 48 | 0.4480 | **0.2860** | **0.4374** | 0.2863 |
| | 96 | 0.3830 | **0.2590** | **0.3757** | 0.2608 |
| | 144 | 0.3800 | **0.2580** | **0.3781** | 0.2615 |
| | 192 | 0.3870 | **0.2620** | **0.3809** | 0.2644 |
| ETTm1 | 48 | **0.2770** | **0.3180** | 0.2820 | 0.3319 |
| | 96 | **0.2880** | **0.3280** | 0.2882 | 0.3425 |
| | 144 | **0.3080** | **0.3440** | 0.3106 | 0.3581 |
| | 192 | **0.3220** | **0.3530** | 0.3233 | 0.3646 |
| ETTm2 | 48 | **0.1330** | **0.2210** | 0.1350 | 0.2309 |
| | 96 | 0.1690 | **0.2480** | **0.1676** | 0.2548 |
| | 144 | 0.2030 | **0.2710** | **0.1991** | 0.2774 |
| | 192 | 0.2240 | **0.2890** | **0.2205** | 0.2924 |
| Weather | 48 | 0.1380 | **0.1680** | **0.1376** | 0.1774 |
| | 96 | 0.1570 | **0.1920** | **0.1551** | 0.1998 |
| | 144 | 0.1740 | **0.2100** | **0.1733** | 0.2201 |
| | 192 | 0.1910 | **0.2260** | **0.1909** | 0.2392 |

## A.6. Stability of Results

To assess the robustness of our main results on the standard multivariate forecasting benchmark (Table 1), we evaluate the stability of each metric by computing its standard deviation across five independent random seeds. The detailed results are presented in Table 10. As shown, the standard deviations are consistently low across most datasets, indicating that *DeepEDM* yields stable and reliable forecasts. As expected, the ILI dataset, being much smaller, exhibits relatively higher variance.

## A.7. Additional Details on Synthetic Data Experiments

**Data Generation:** To systematically analyze model performance under deterministic yet unpredictable systems, we generate synthetic datasets for both *chaotic* and *non-chaotic* dynamical systems. Non-chaotic systems exhibit predictable behavior, where small variations in initial conditions result in only minor deviations in long-term trajectories. In contrast, chaotic systems, despite being governed by deterministic rules, exhibit extreme sensitivity to initial conditions, leading to exponentially diverging trajectories over time. A canonical example of deterministic chaos, the Lorenz system, is governed by a set of nonlinear differential equations that give rise to a strange attractor, characterized by a series of bifurcations and highly sensitive trajectory evolution (Figure 5, middle row).

To capture distinct dynamical regimes of the Lorenz system, we generate two configurations: (i) *Chaotic behavior*: $\sigma = 10.0$, $\rho = 28.0$, $\beta = 2.667$ and initial conditions: $(0.0, 1.0, 1.05)$ (ii) *Non-chaotic behavior:* $\sigma = 10.0$, $\rho = 9$, $\beta = 2.667$, and initial conditions: $(10.0, 10.0, 10.0)$. For the Rössler system, we generate one chaotic configuration using $a = 0.2, b = 0.2, c = 5.7$ and initial conditions: $(1., 1., 1.)$.

*Table 10.* Standard deviation of the main results (Table 1) over five seeds across different forecast horizons: $H \in \{48, 96, 144, 192\}$ for all datasets, except ILI, where $H \in \{24, 36, 48, 60\}$. The results demonstrate stability, with consistently low standard deviations across datasets. However, the ILI dataset, being significantly smaller, naturally exhibits relatively higher variance, particularly for shorter forecast horizons.

| Dataset | Forecast Length ($H$) | | | | | | | |
|---|---|---|---|---|---|---|---|---|
| | $H_0$ | | $H_1$ | | $H_2$ | | $H_3$ | |
| | $\sigma_{MSE}$ | $\sigma_{MAE}$ | $\sigma_{MSE}$ | $\sigma_{MAE}$ | $\sigma_{MSE}$ | $\sigma_{MAE}$ | $\sigma_{MSE}$ | $\sigma_{MAE}$ |
| ECL | 0.0007 | 0.0006 | 0.0006 | 0.0004 | 0.0008 | 0.0008 | 0.0003 | 0.0005 |
| ETTh1 | 0.0015 | 0.0009 | 0.0038 | 0.0008 | 0.0017 | 0.0006 | 0.0055 | 0.0050 |
| ETTh2 | 0.0018 | 0.0009 | 0.0013 | 0.0012 | 0.0081 | 0.0033 | 0.0077 | 0.0036 |
| ETTm1 | 0.0010 | 0.0005 | 0.0006 | 0.0005 | 0.0020 | 0.0005 | 0.0014 | 0.0008 |
| ETTm2 | 0.0004 | 0.0003 | 0.0010 | 0.0005 | 0.0028 | 0.0016 | 0.0023 | 0.0013 |
| Traffic | 0.0033 | 0.0008 | 0.0007 | 0.0009 | 0.0008 | 0.0007 | 0.0012 | 0.0012 |
| Exchange | 0.0006 | 0.0013 | 0.0021 | 0.0020 | 0.0038 | 0.0036 | 0.0041 | 0.0026 |
| ILI | 0.1383 | 0.0272 | 0.0642 | 0.0104 | 0.0380 | 0.0103 | 0.0621 | 0.0141 |
| Weather | 0.0007 | 0.0009 | 0.0003 | 0.0003 | 0.0005 | 0.0003 | 0.0005 | 0.0003 |

The initial conditions were selected to ensure trajectories remain well-defined and not excessively perturbed even under the highest noise settings. Further to systematically evaluate model performance in presence of noise, we simulate noisy conditions by introducing Gaussian noise $N(0, \sigma_{noise}^2)$, with $\sigma_{noise} \in \{0.0, 0.5, 1.0, 1.5, 2.0, 2.5\}$ with higher $\sigma_{noise}$ denoting higher levels of noise. This results in a total of 18 synthetic datasets (3 systems × 6 noise levels). The underlying dynamical systems and noise levels are illustrated in Figure 5.

**Experimental Setup:** Each synthetic dataset is divided into sequential non-overlapping training, validation, and testing splits. The models are trained on their respective training sets and evaluated on the test sets, with validation sets used for early stopping to mitigate overfitting.

All learning-based models are trained to forecast a fixed number of future steps (96 steps) based on a fixed lookback window (192 steps). The performance of each model is assessed based on its ability to forecast accurately over varying lengths ($p$) and under different noise conditions. Specifically, while models are trained with a forecast length of 48 steps, only the first $p$ steps of each forecast are considered during testing. This evaluation strategy ensures a consistent and unbiased comparison across different prediction lengths and models.

**Results:** Table 11 details the quantitative results of our experiments on synthetic datasets under varying noise levels ($\sigma_{noise}$) and prediction horizons ($H$) across three dynamical systems. *DeepEDM* consistently delivers the lowest MSE and MAE, demonstrating superior forecast performance, particularly in noisy and chaotic regimes.

*Chaotic datasets:* Chaotic regimes pose significant challenges for long-term forecasting due to their inherent complexity, leading to relatively high errors. Despite these challenges, *DeepEDM* handles forecasting far more effectively than the baseline methods. At $\sigma = 2.5$ and $H = 48$, its MSE (17.267) is 40% lower than *Koopa* (28.804), 45% below *iTransformer* (31.599), and 60% below *Simplex* (43.548). This advantage is also evident in the no-noise regime ($\sigma_{noise} = 0$, $H = 48$), where *DeepEDM* achieves an MSE of 10.467—outperforming *Koopa* (18.978) by 44.85%, *iTransformer* (18.531) by 43.52%, and *Simplex* (30.985) by 66.22%. *DeepEDM* also demonstrates consistent superiority on the Rössler system, particularly in terms of MAE, further reinforcing its robustness against noise and its ability to model complex nonlinear dynamics.

*Non-chaotic datasets:* In the simpler non-chaotic setting with no noise, all the baselines perform comparably, however as the noise level increases *DeepEDM* still maintains a competitive edge. For instant, at $\sigma = 2.0_{noise}$ and $H = 48$, it achieves an MSE of 0.048—18% lower than *Koopa* (0.059) and 6% lower than *iTransformer* (0.051), while *Simplex* deteriorates drastically to 3.921.

In summary, across all three systems, *DeepEDM* outperforms both classical EDM methods and learning based baselines, showcasing its resilience under noise and across varying prediction horizons.

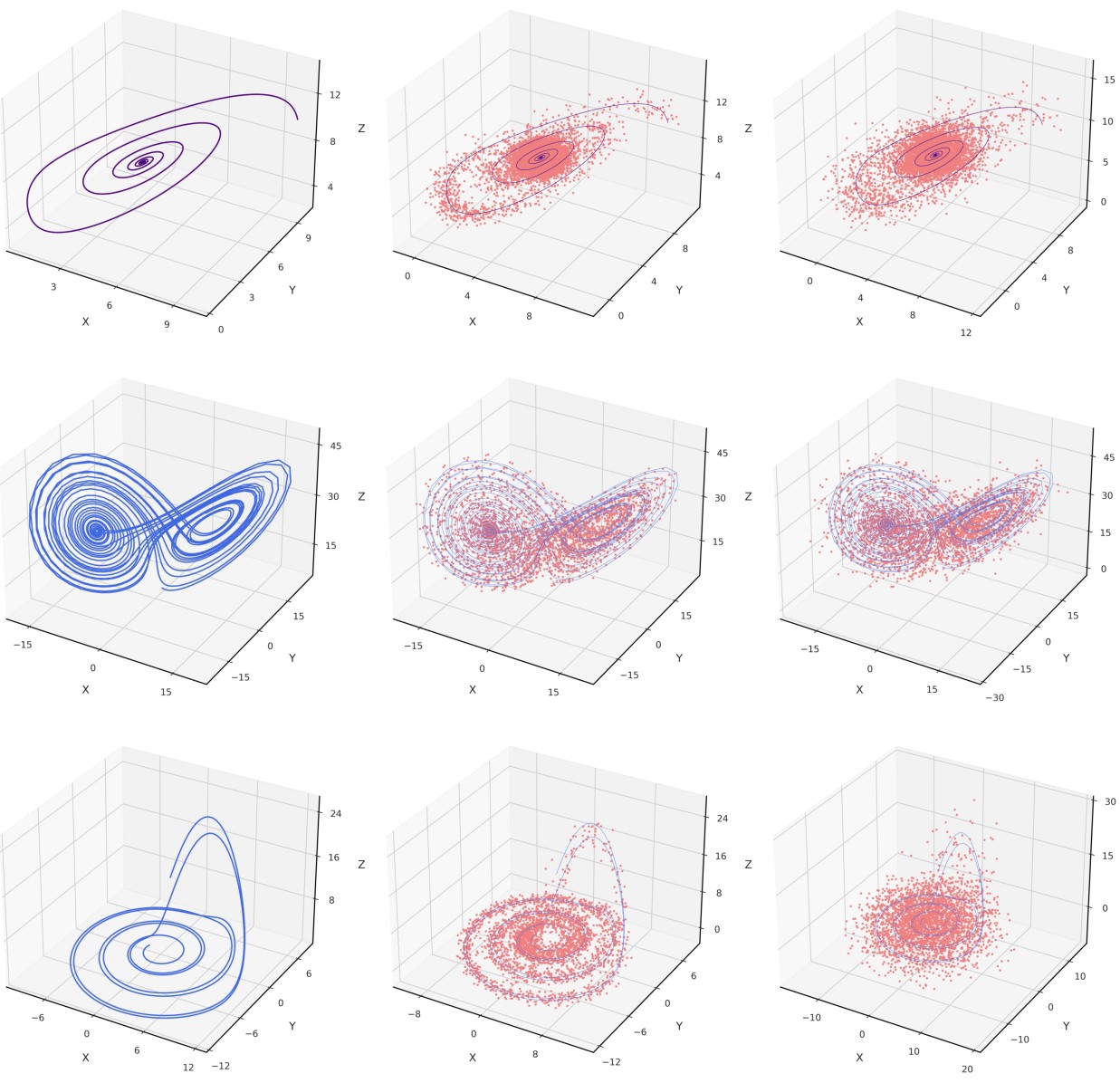

*Figure 5.* Visualization of the synthetic datasets: Non-chaotic Lorenz (top row), chaotic Lorenz (middle row), and chaotic Rössler (bottom row) systems. As the noise level increases (from left to right), forecasting future states becomes progressively more challenging. Pink dots indicate the new attractor under the current regime, while the light blue denotes the original attractor for reference.

Table 11. Multivariate forecasting results with prediction lengths $H \in [1, 5, 15, 48]$ for multiple dynamical systems. The amount of Gaussian noise added is linearly increased by varying the standard deviation.

| Models | | DeepEDM | | | | | | | | Koopa | | | | | | | | iTransformer | | | | | | | | Simplex | | | | | | | |
|---|---|---|---|---|---|---|---|---|---|---|---|---|---|---|---|---|---|---|---|---|---|---|---|---|---|---|---|---|---|---|---|---|---|
| H | | 1 | | 5 | | 15 | | 48 | | 1 | | 5 | | 15 | | 48 | | 1 | | 5 | | 15 | | 48 | | 1 | | 5 | | 15 | | 48 | |
| σ | | MSE | MAE | MSE | MAE | MSE | MAE | MSE | MAE | MSE | MAE | MSE | MAE | MSE | MAE | MSE | MAE | MSE | MAE | MSE | MAE | MSE | MAE | MSE | MAE | MSE | MAE | MSE | MAE | MSE | MAE | MSE | MAE |
| **Non Chaotic Lorenz** | 0.0 | 0.000 | 0.000 | 0.000 | 0.000 | 0.000 | 0.000 | 0.000 | 0.000 | 0.000 | 0.000 | 0.000 | 0.000 | 0.000 | 0.000 | 0.000 | 0.000 | 0.000 | 0.005 | 0.000 | 0.004 | 0.000 | 0.004 | 0.000 | 0.005 | 0.000 | 0.000 | 0.000 | 0.000 | 0.000 | 0.000 | 0.000 | 0.000 |
| | 0.5 | 0.004 | 0.050 | 0.004 | 0.050 | 0.004 | 0.050 | 0.005 | 0.053 | 0.004 | 0.052 | 0.004 | 0.052 | 0.005 | 0.054 | 0.007 | 0.061 | 0.005 | 0.054 | 0.005 | 0.057 | 0.005 | 0.058 | 0.006 | 0.062 | 0.245 | 0.395 | 0.245 | 0.395 | 0.245 | 0.395 | 0.245 | 0.395 |
| | 1.0 | 0.013 | 0.091 | 0.014 | 0.092 | 0.014 | 0.093 | 0.017 | 0.099 | 0.014 | 0.092 | 0.014 | 0.092 | 0.015 | 0.095 | 0.019 | 0.106 | 0.015 | 0.098 | 0.016 | 0.100 | 0.016 | 0.102 | 0.019 | 0.110 | 0.981 | 0.790 | 0.981 | 0.790 | 0.981 | 0.790 | 0.980 | 0.789 |
| | 1.5 | 0.025 | 0.126 | 0.025 | 0.126 | 0.026 | 0.127 | 0.030 | 0.135 | 0.030 | 0.136 | 0.030 | 0.135 | 0.032 | 0.139 | 0.040 | 0.153 | 0.027 | 0.131 | 0.028 | 0.133 | 0.030 | 0.136 | 0.034 | 0.145 | 2.208 | 1.185 | 2.208 | 1.185 | 2.208 | 1.185 | 2.206 | 1.184 |
| | 2.0 | 0.040 | 0.159 | 0.041 | 0.161 | 0.042 | 0.161 | 0.048 | 0.171 | 0.047 | 0.167 | 0.046 | 0.166 | 0.049 | 0.171 | 0.059 | 0.186 | 0.042 | 0.164 | 0.044 | 0.167 | 0.046 | 0.170 | 0.051 | 0.179 | 3.925 | 1.579 | 3.925 | 1.580 | 3.926 | 1.580 | 3.921 | 1.578 |
| | 2.5 | 0.059 | 0.192 | 0.061 | 0.195 | 0.061 | 0.195 | 0.069 | 0.206 | 0.064 | 0.196 | 0.066 | 0.198 | 0.069 | 0.202 | 0.081 | 0.216 | 0.058 | 0.194 | 0.061 | 0.197 | 0.063 | 0.200 | 0.069 | 0.209 | 6.132 | 1.974 | 6.133 | 1.974 | 6.134 | 1.974 | 6.127 | 1.973 |
| **Chaotic Lorenz** | 0.0 | 0.056 | 0.130 | 0.360 | 0.320 | 2.414 | 0.763 | 10.476 | 1.613 | 1.772 | 0.912 | 2.276 | 0.986 | 6.132 | 1.484 | 18.978 | 2.660 | 3.064 | 1.229 | 3.995 | 1.344 | 8.018 | 1.753 | 18.531 | 2.625 | 8.696 | 2.048 | 12.259 | 2.458 | 20.885 | 3.221 | 30.905 | 3.952 |
| | 0.5 | 0.255 | 0.364 | 0.797 | 0.556 | 3.642 | 1.006 | 12.632 | 1.823 | 2.108 | 1.069 | 2.797 | 1.191 | 7.024 | 1.687 | 19.774 | 2.767 | 3.409 | 1.331 | 4.374 | 1.446 | 8.457 | 1.849 | 19.097 | 2.712 | 9.158 | 2.143 | 12.764 | 2.545 | 21.529 | 3.303 | 31.806 | 4.041 |
| | 1.0 | 0.597 | 0.559 | 1.412 | 0.758 | 4.718 | 1.184 | 13.598 | 1.937 | 2.976 | 1.327 | 3.817 | 1.454 | 8.644 | 1.967 | 21.518 | 2.987 | 4.437 | 1.581 | 5.469 | 1.694 | 9.697 | 2.084 | 20.535 | 2.917 | 10.406 | 2.357 | 14.094 | 2.741 | 23.122 | 3.482 | 33.864 | 4.226 |
| | 1.5 | 0.979 | 0.733 | 1.937 | 0.932 | 5.444 | 1.325 | 14.675 | 2.017 | 4.682 | 1.694 | 5.833 | 1.835 | 11.317 | 2.343 | 24.270 | 3.297 | 6.140 | 1.909 | 7.232 | 2.015 | 11.645 | 2.391 | 22.689 | 3.184 | 12.318 | 2.633 | 16.095 | 2.994 | 25.392 | 3.712 | 36.597 | 4.455 |
| | 2.0 | 1.427 | 0.890 | 2.549 | 1.093 | 6.421 | 1.505 | 15.821 | 2.174 | 7.089 | 2.092 | 8.339 | 2.228 | 13.968 | 2.698 | 26.811 | 3.592 | 8.456 | 2.272 | 9.608 | 2.374 | 14.250 | 2.738 | 25.467 | 3.486 | 14.838 | 2.946 | 18.696 | 3.283 | 28.233 | 3.974 | 39.844 | 4.711 |
| | 2.5 | 1.913 | 1.024 | 3.212 | 1.228 | 7.555 | 1.653 | 17.267 | 2.338 | 9.441 | 2.425 | 10.867 | 2.571 | 17.055 | 3.046 | 30.383 | 3.915 | 11.328 | 2.649 | 12.551 | 2.748 | 17.449 | 3.104 | 28.804 | 3.807 | 17.938 | 3.281 | 21.864 | 3.595 | 31.599 | 4.258 | 43.548 | 4.982 |
| **Chaotic Rössler** | 0.0 | 0.028 | 0.067 | 0.030 | 0.069 | 0.033 | 0.074 | 0.291 | 0.132 | 0.048 | 0.127 | 0.042 | 0.121 | 0.035 | 0.114 | 0.299 | 0.170 | 0.089 | 0.178 | 0.085 | 0.171 | 0.107 | 0.171 | 0.684 | 0.225 | 1.029 | 0.647 | 1.096 | 0.668 | 1.359 | 0.724 | 2.388 | 0.897 |
| | 0.5 | 0.054 | 0.152 | 0.064 | 0.164 | 0.101 | 0.202 | 0.783 | 0.332 | 0.119 | 0.252 | 0.114 | 0.247 | 0.144 | 0.271 | 0.717 | 0.402 | 0.161 | 0.293 | 0.168 | 0.303 | 0.198 | 0.314 | 0.787 | 0.400 | 1.457 | 0.889 | 1.528 | 0.906 | 1.811 | 0.959 | 3.000 | 1.137 |
| | 1.0 | 0.127 | 0.235 | 0.151 | 0.251 | 0.218 | 0.284 | 1.066 | 0.429 | 0.141 | 0.287 | 0.154 | 0.298 | 0.213 | 0.339 | 0.959 | 0.513 | 0.245 | 0.386 | 0.257 | 0.393 | 0.304 | 0.411 | 0.893 | 0.516 | 2.566 | 1.240 | 2.643 | 1.255 | 2.954 | 1.302 | 4.313 | 1.475 |
| | 1.5 | 0.222 | 0.323 | 0.260 | 0.342 | 0.372 | 0.384 | 1.320 | 0.549 | 0.264 | 0.387 | 0.271 | 0.391 | 0.349 | 0.430 | 1.312 | 0.635 | 0.346 | 0.457 | 0.365 | 0.465 | 0.424 | 0.490 | 1.002 | 0.611 | 4.273 | 1.625 | 4.354 | 1.638 | 4.694 | 1.682 | 6.191 | 1.846 |
| | 2.0 | 0.303 | 0.385 | 0.342 | 0.402 | 0.478 | 0.446 | 1.540 | 0.630 | 0.357 | 0.461 | 0.380 | 0.472 | 0.494 | 0.521 | 1.510 | 0.732 | 0.456 | 0.527 | 0.469 | 0.531 | 0.545 | 0.561 | 1.131 | 0.691 | 6.547 | 2.024 | 6.634 | 2.036 | 7.002 | 2.079 | 8.605 | 2.233 |
| | 2.5 | 0.453 | 0.481 | 0.519 | 0.504 | 0.736 | 0.561 | 1.913 | 0.765 | 0.428 | 0.492 | 0.461 | 0.507 | 0.590 | 0.560 | 1.599 | 0.801 | 0.641 | 0.619 | 0.648 | 0.620 | 0.743 | 0.651 | 1.368 | 0.791 | 9.369 | 2.429 | 9.461 | 2.440 | 9.856 | 2.481 | 11.534 | 2.625 |

*Table 12.* Full ablation table

| Dataset | | Linear | | MLP | | MLP+EDM | | Full Model | |
|---|---|---|---|---|---|---|---|---|---|
| Metric | | MSE | MAE | MSE | MAE | MSE | MAE | MSE | MAE |
| ECL | 48 | $0.1980_{\pm0.0001}$ | $0.2704_{\pm0.0004}$ | $0.1889_{\pm0.0000}$ | $0.2673_{\pm0.0001}$ | $\mathbf{0.1591}_{\pm0.0001}$ | $\mathbf{0.2463}_{\pm0.0002}$ | $0.1607_{\pm0.0011}$ | $0.2470_{\pm0.0009}$ |
| | 96 | $0.1534_{\pm0.0002}$ | $0.2463_{\pm0.0003}$ | $0.1511_{\pm0.0028}$ | $0.2478_{\pm0.0024}$ | $0.1380_{\pm0.0001}$ | $0.2322_{\pm0.0001}$ | $\mathbf{0.1377}_{\pm0.0008}$ | $\mathbf{0.2317}_{\pm0.0006}$ |
| | 144 | $0.1526_{\pm0.0001}$ | $0.2461_{\pm0.0002}$ | $0.1531_{\pm0.0016}$ | $0.2494_{\pm0.0019}$ | $0.1466_{\pm0.0002}$ | $0.2393_{\pm0.0003}$ | $\mathbf{0.1459}_{\pm0.0008}$ | $\mathbf{0.2392}_{\pm0.0009}$ |
| | 192 | $0.1545_{\pm0.0001}$ | $0.2486_{\pm0.0002}$ | $0.1533_{\pm0.0008}$ | $0.2483_{\pm0.0014}$ | $0.1528_{\pm0.0008}$ | $0.2458_{\pm0.0005}$ | $\mathbf{0.1507}_{\pm0.0003}$ | $\mathbf{0.2438}_{\pm0.0007}$ |
| ETTh1 | 48 | $0.3412_{\pm0.0012}$ | $0.3630_{\pm0.0015}$ | $0.3369_{\pm0.0006}$ | $0.3654_{\pm0.0003}$ | $0.3280_{\pm0.0006}$ | $0.3599_{\pm0.0002}$ | $\mathbf{0.3236}_{\pm0.0012}$ | $\mathbf{0.3566}_{\pm0.0004}$ |
| | 96 | $0.3759_{\pm0.0014}$ | $0.3866_{\pm0.0012}$ | $0.3712_{\pm0.0007}$ | $0.3861_{\pm0.0007}$ | $0.3687_{\pm0.0034}$ | $0.3873_{\pm0.0007}$ | $\mathbf{0.3622}_{\pm0.0033}$ | $\mathbf{0.3831}_{\pm0.0004}$ |
| | 144 | $0.3953_{\pm0.0009}$ | $0.4005_{\pm0.0009}$ | $0.3961_{\pm0.0011}$ | $0.4031_{\pm0.0014}$ | $0.3981_{\pm0.0041}$ | $0.4022_{\pm0.0019}$ | $\mathbf{0.3885}_{\pm0.0026}$ | $\mathbf{0.3977}_{\pm0.0006}$ |
| | 192 | $0.4096_{\pm0.0009}$ | $0.4125_{\pm0.0013}$ | $0.4085_{\pm0.0010}$ | $\mathbf{0.4113}_{\pm0.0015}$ | $0.4180_{\pm0.0033}$ | $0.4261_{\pm0.0017}$ | $\mathbf{0.4064}_{\pm0.0074}$ | $0.4213_{\pm0.0070}$ |
| ETTh2 | 48 | $0.2258_{\pm0.0003}$ | $0.2913_{\pm0.0002}$ | $\mathbf{0.2237}_{\pm0.0009}$ | $0.2907_{\pm0.0003}$ | $0.2265_{\pm0.0022}$ | $0.2886_{\pm0.0010}$ | $0.2256_{\pm0.0021}$ | $\mathbf{0.2875}_{\pm0.0011}$ |
| | 96 | $0.2838_{\pm0.0002}$ | $0.3331_{\pm0.0002}$ | $\mathbf{0.2807}_{\pm0.0002}$ | $0.3310_{\pm0.0004}$ | $0.2935_{\pm0.0040}$ | $0.3367_{\pm0.0023}$ | $0.2882_{\pm0.0013}$ | $0.3329_{\pm0.0017}$ |
| | 144 | $0.3202_{\pm0.0008}$ | $0.3580_{\pm0.0003}$ | $\mathbf{0.3161}_{\pm0.0020}$ | $\mathbf{0.3546}_{\pm0.0011}$ | $0.3317_{\pm0.0022}$ | $0.3676_{\pm0.0018}$ | $0.3195_{\pm0.0009}$ | $0.3604_{\pm0.0016}$ |
| | 192 | $0.3505_{\pm0.0017}$ | $0.3787_{\pm0.0008}$ | $\mathbf{0.3434}_{\pm0.0035}$ | $\mathbf{0.3745}_{\pm0.0015}$ | $0.3551_{\pm0.0084}$ | $0.3803_{\pm0.0033}$ | $0.3483_{\pm0.0099}$ | $0.3757_{\pm0.0047}$ |
| ETTm1 | 48 | $0.3045_{\pm0.0006}$ | $0.3346_{\pm0.0003}$ | $0.3045_{\pm0.0004}$ | $0.3346_{\pm0.0004}$ | $\mathbf{0.2766}_{\pm0.0009}$ | $\mathbf{0.3178}_{\pm0.0004}$ | $0.2768_{\pm0.0007}$ | $0.3184_{\pm0.0007}$ |
| | 96 | $0.3043_{\pm0.0015}$ | $0.3377_{\pm0.0011}$ | $0.3035_{\pm0.0010}$ | $0.3367_{\pm0.0004}$ | $0.2924_{\pm0.0017}$ | $0.3305_{\pm0.0010}$ | $\mathbf{0.2885}_{\pm0.0003}$ | $\mathbf{0.3279}_{\pm0.0006}$ |
| | 144 | $0.3218_{\pm0.0002}$ | $0.3492_{\pm0.0001}$ | $0.3129_{\pm0.0009}$ | $0.3470_{\pm0.0002}$ | $0.3106_{\pm0.0016}$ | $0.3463_{\pm0.0010}$ | $\mathbf{0.3070}_{\pm0.0024}$ | $\mathbf{0.3437}_{\pm0.0007}$ |
| | 192 | $0.3364_{\pm0.0002}$ | $0.3581_{\pm0.0008}$ | $0.3282_{\pm0.0004}$ | $0.3583_{\pm0.0002}$ | $0.3279_{\pm0.0012}$ | $0.3566_{\pm0.0004}$ | $\mathbf{0.3213}_{\pm0.0008}$ | $\mathbf{0.3527}_{\pm0.0011}$ |
| ETTm2 | 48 | $0.1439_{\pm0.0002}$ | $0.2334_{\pm0.0000}$ | $0.1437_{\pm0.0003}$ | $0.2335_{\pm0.0001}$ | $0.1338_{\pm0.0006}$ | $0.2221_{\pm0.0008}$ | $\mathbf{0.1330}_{\pm0.0006}$ | $\mathbf{0.2210}_{\pm0.0005}$ |
| | 96 | $0.1718_{\pm0.0003}$ | $0.2510_{\pm0.0001}$ | $0.1717_{\pm0.0002}$ | $0.2510_{\pm0.0001}$ | $0.1710_{\pm0.0028}$ | $0.2502_{\pm0.0022}$ | $\mathbf{0.1686}_{\pm0.0013}$ | $\mathbf{0.2478}_{\pm0.0006}$ |
| | 144 | $\mathbf{0.1993}_{\pm0.0004}$ | $0.2708_{\pm0.0001}$ | $\mathbf{0.1993}_{\pm0.0009}$ | $\mathbf{0.2695}_{\pm0.0003}$ | $0.2011_{\pm0.0042}$ | $0.2711_{\pm0.0027}$ | $0.2020_{\pm0.0042}$ | $0.2714_{\pm0.0025}$ |
| | 192 | $0.2201_{\pm0.0002}$ | $0.2855_{\pm0.0002}$ | $\mathbf{0.2198}_{\pm0.0002}$ | $\mathbf{0.2852}_{\pm0.0001}$ | $0.2261_{\pm0.0040}$ | $0.2908_{\pm0.0029}$ | $0.2234_{\pm0.0027}$ | $0.2886_{\pm0.0016}$ |
| Traffic | 48 | $0.6978_{\pm0.0005}$ | $0.4131_{\pm0.0004}$ | $0.5621_{\pm0.0012}$ | $0.3636_{\pm0.0003}$ | $\mathbf{0.4374}_{\pm0.0053}$ | $0.2863_{\pm0.0013}$ | $0.4501_{\pm0.0008}$ | $\mathbf{0.2859}_{\pm0.0011}$ |
| | 96 | $0.4508_{\pm0.0006}$ | $0.2985_{\pm0.0013}$ | $0.4241_{\pm0.0024}$ | $0.2974_{\pm0.0023}$ | $0.3757_{\pm0.0015}$ | $0.2608_{\pm0.0014}$ | $\mathbf{0.3828}_{\pm0.0006}$ | $\mathbf{0.2596}_{\pm0.0004}$ |
| | 144 | $0.4300_{\pm0.0007}$ | $0.2912_{\pm0.0015}$ | $0.4112_{\pm0.0013}$ | $0.2871_{\pm0.0012}$ | $\mathbf{0.3781}_{\pm0.0014}$ | $0.2615_{\pm0.0016}$ | $0.3804_{\pm0.0011}$ | $\mathbf{0.2582}_{\pm0.0009}$ |
| | 192 | $0.4219_{\pm0.0004}$ | $0.2876_{\pm0.0012}$ | $0.4111_{\pm0.0047}$ | $0.2933_{\pm0.0040}$ | $\mathbf{0.3809}_{\pm0.0019}$ | $0.2644_{\pm0.0020}$ | $0.3870_{\pm0.0017}$ | $\mathbf{0.2616}_{\pm0.0018}$ |
| Exchange | 48 | $0.0430_{\pm0.0003}$ | $0.1428_{\pm0.0006}$ | $0.0423_{\pm0.0001}$ | $\mathbf{0.1409}_{\pm0.0000}$ | $0.0427_{\pm0.0003}$ | $0.1422_{\pm0.0007}$ | $\mathbf{0.0422}_{\pm0.0003}$ | $0.1411_{\pm0.0006}$ |
| | 96 | $\mathbf{0.0841}_{\pm0.0004}$ | $\mathbf{0.2018}_{\pm0.0006}$ | $0.0853_{\pm0.0018}$ | $0.2029_{\pm0.0021}$ | $0.0853_{\pm0.0014}$ | $0.2037_{\pm0.0010}$ | $0.0870_{\pm0.0024}$ | $0.2044_{\pm0.0023}$ |
| | 144 | $\mathbf{0.1301}_{\pm0.0000}$ | $\mathbf{0.2526}_{\pm0.0002}$ | $0.1314_{\pm0.0010}$ | $0.2550_{\pm0.0011}$ | $0.1370_{\pm0.0062}$ | $0.2603_{\pm0.0056}$ | $0.1319_{\pm0.0053}$ | $0.2546_{\pm0.0049}$ |
| | 192 | $0.1818_{\pm0.0017}$ | $0.3018_{\pm0.0017}$ | $0.1849_{\pm0.0047}$ | $0.3061_{\pm0.0025}$ | $0.1838_{\pm0.0038}$ | $0.3053_{\pm0.0023}$ | $\mathbf{0.1751}_{\pm0.0024}$ | $\mathbf{0.2989}_{\pm0.0004}$ |
| ILI | 24 | $2.2266_{\pm0.0111}$ | $0.9375_{\pm0.0049}$ | $2.2114_{\pm0.0117}$ | $0.8901_{\pm0.0066}$ | $1.8514_{\pm0.1304}$ | $0.8193_{\pm0.0150}$ | $\mathbf{1.7489}_{\pm0.1807}$ | $\mathbf{0.7896}_{\pm0.0362}$ |
| | 36 | $2.0815_{\pm0.0098}$ | $0.9264_{\pm0.0023}$ | $1.9350_{\pm0.0201}$ | $0.8443_{\pm0.0107}$ | $\mathbf{1.5775}_{\pm0.0185}$ | $\mathbf{0.7606}_{\pm0.0049}$ | $1.6405_{\pm0.0709}$ | $0.7690_{\pm0.0157}$ |
| | 48 | $1.8621_{\pm0.0090}$ | $0.9020_{\pm0.0011}$ | $1.6525_{\pm0.0327}$ | $0.8133_{\pm0.0130}$ | $1.6184_{\pm0.0798}$ | $\mathbf{0.7868}_{\pm0.0169}$ | $\mathbf{1.6149}_{\pm0.0572}$ | $0.7880_{\pm0.0138}$ |
| | 60 | $1.9258_{\pm0.1907}$ | $0.9425_{\pm0.0583}$ | $2.1317_{\pm0.2852}$ | $0.9979_{\pm0.1007}$ | $\mathbf{1.6955}_{\pm0.0934}$ | $\mathbf{0.8241}_{\pm0.0290}$ | $1.7072_{\pm0.0921}$ | $0.8274_{\pm0.0209}$ |
| Weather | 48 | $0.1665_{\pm0.0012}$ | $0.1919_{\pm0.0029}$ | $0.1618_{\pm0.0012}$ | $0.1870_{\pm0.0013}$ | $0.1396_{\pm0.0020}$ | $0.1705_{\pm0.0028}$ | $\mathbf{0.1382}_{\pm0.0010}$ | $\mathbf{0.1683}_{\pm0.0013}$ |
| | 96 | $0.1938_{\pm0.0006}$ | $0.2218_{\pm0.0014}$ | $0.1931_{\pm0.0004}$ | $0.2196_{\pm0.0005}$ | $\mathbf{0.1566}_{\pm0.0004}$ | $\mathbf{0.1917}_{\pm0.0005}$ | $0.1571_{\pm0.0001}$ | $\mathbf{0.1917}_{\pm0.0001}$ |
| | 144 | $0.2033_{\pm0.0004}$ | $0.2341_{\pm0.0006}$ | $0.1863_{\pm0.0009}$ | $0.2214_{\pm0.0008}$ | $0.1752_{\pm0.0006}$ | $0.2105_{\pm0.0006}$ | $\mathbf{0.1739}_{\pm0.0003}$ | $\mathbf{0.2093}_{\pm0.0002}$ |
| | 192 | $0.2185_{\pm0.0005}$ | $0.2518_{\pm0.0014}$ | $0.2182_{\pm0.0006}$ | $0.2512_{\pm0.0001}$ | $0.1924_{\pm0.0015}$ | $0.2281_{\pm0.0012}$ | $\mathbf{0.1913}_{\pm0.0007}$ | $\mathbf{0.2263}_{\pm0.0003}$ |
| #Improvements #Degradations | | Baseline | | 55 17 | | 52 20 | | 54 18 | |

