# OpenReview forum: "LETS Forecast: Learning Embedology for Time Series Forecasting"
_ICML.cc/2025/Conference — ICML 2025 poster_

### Official Review · Reviewer_XFih · 2025-03-02

**Overall Recommendation:** 3

**Summary:**

In this paper, the authors introduce DeepEDM, a framework that extends Empirical Dynamical Modeling (EDM) to learn complex latent non-linear dynamics from observed time series for improved forecasting. Building upon Takens’ theorem, DeepEDM first constructs time-delayed embeddings of the input data and then projects these embeddings into a high-dimensional latent space. Next, it employs kernel regression on this latent representation, and uses a multilayer perceptron (MLP) to transform the resulting predictions back to the original time-series space.

According to the authors, DeepEDM offers three key advantages over existing methods:
1. It remains robust even in the presence of noisy observations.
2. It trains a unified parametric model that readily adapts to new time-series data.
3. It combines the rigor of dynamical systems with the flexibility and scalability through its deep learning–based architecture.

In addition to describing fundamental concepts in empirical dynamical modeling and reviewing relevant literature on deep learning for time-series forecasting, the paper also draws connections between DeepEDM and the Attention mechanism. Extensive experiments on both simulated and real-world multivariate forecasting benchmarks show that DeepEDM performs competitively with state-of-the-art approaches, highlighting its potential for robust and adaptable time-series forecasting.

**Claims And Evidence:**

Authors claim that DeepEDM:
1. remains robust even in the presence of noisy observations.
2. trains a unified parametric model that readily adapts to new time-series data.
3. combines the rigor of dynamical systems with the flexibility and scalability through its deep learning–based architecture.

Overall, the paper effectively supports its first claim—that DeepEDM remains robust in the presence of noisy observations, by demonstrating strong performance in both synthetic noise-injected scenarios and on real-world datasets against various state-of-the-art models.

The second claim, involving a “unified parametric model” that readily adapts to new time-series data, is the least substantiated: after introducing the idea, the paper does not provide further elaboration or experiments to confirm how DeepEDM generalizes to new time series. Consequently, while the framework appears promising, further clarity is needed on the role and validation of the latent dynamics, as well as concrete evidence of adaptation to new data.

It also partially supports its third claim of merging dynamical systems rigor with deep learning flexibility, as the authors present a novel architecture grounded in Takens’ theorem and empirical dynamical modeling. While the overall design is innovative, key details are missing about the latent-projection step. Specifically, it is unclear whether the “latent dynamics” it learns genuinely reflect the underlying system or directly improve the forecast. The paper does not delve into interpretability or show whether these latent dynamics match “true” hidden states in controlled scenarios (beyond broad performance metrics).

**Essential References Not Discussed:**

The paper covers most of the relevant literature on empirical dynamical modeling (EDM) and deep time-series forecasting. However, despite DeepEDM’s direct connection to chaotic time series prediction, the authors do not provide a literature review of this field. Foundational work, such as Farmer & Sidorowich (1987), established key principles for short-term chaotic forecasting using time-delay embeddings, which are directly relevant to DeepEDM’s methodology. More recent studies have extended these ideas using neural networks, including:

1. Karunasinghe, D. S., & Liong, S. Y. (2006). Chaotic time series prediction with a global model: Artificial neural network. Journal of Hydrology, 323(1-4), 92-105.
2. Li, Decai, Min Han, and Jun Wang. "Chaotic time series prediction based on a novel robust echo state network." IEEE Transactions on Neural Networks and Learning Systems 23.5 (2012): 787-799.

Including a discussion of this literature would better position DeepEDM in the context of prior neural-based approaches to chaotic forecasting and clarify its novelty in comparison to existing methods.

**Experimental Designs Or Analyses:**

In the simulation study, authors inject Gaussian noise into well-known chaotic systems (e.g., Lorenz, Rössler) to test the model’s robustness; this is a reasonable to demonstrate DeepEDM's resilience to noise. However, as previously noted, it only compares DeepEDM with a limited set of methods (mostly EDM-based). This narrow comparison leaves open the question of how DeepEDM would fare against a broader set of state-of-the-art models in similarly controlled, noise-injected scenarios.

**Methods And Evaluation Criteria:**

Yes, the paper’s methods and chosen benchmarks do make sense for time-series forecasting, as it uses standard datasets and builds upon widely studied dynamical systems for its simulations. However, the limited scope of comparisons in the simulated scenarios—focusing mostly on EDM-based approaches such as Simplex, Koopa, and iTransformers—raises some concerns. On the other hand, their comparisons on standard benchmark datasets are reasonably thorough, which partly compensates for the narrower set of baselines in the simulation experiments.

**Other Comments Or Suggestions:**

I have no additional comments.

**Other Strengths And Weaknesses:**

The paper presents a novel combination of Empirical Dynamical Modeling (EDM) and deep learning, which is an original and promising direction for time-series forecasting. However, one key concern is that the model’s performance is primarily evaluated against standard forecasting benchmarks, which may not fully demonstrate its advantage in capturing underlying system dynamics. While these benchmarks are widely used in time-series forecasting, they are not necessarily designed to evaluate methods that explicitly model chaotic or nonlinear dynamical systems. A more extensive comparison against known chaotic time series datasets, beyond the current synthetic experiments in the simulation study, would help clarify whether DeepEDM provides meaningful improvements over deep learning-based methods specifically designed for chaotic time-series prediction.

**Questions For Authors:**

1. How do you validate that the latent space learned by DeepEDM accurately captures the underlying system dynamics?
The paper presents the latent projection as a core component, but it is unclear whether it is reconstructing meaningful system dynamics or simply acting as a high-dimensional feature extractor.

2. Can you clarify what is meant by DeepEDM being a “unified parametric model” that generalizes to new time series?
The introduction suggests that DeepEDM can adapt to new time series data more effectively than EDM, but this claim is not well-supported in the rest of the paper.

3. Why does the paper not reference prior work on chaotic time series forecasting, despite its direct relevance?

**Relation To Broader Scientific Literature:**

The paper appears to be the first to systematically merge Empirical Dynamical Modeling with deep neural architectures for time-series forecasting. While prior work has explored EDM (e.g., Simplex, S-Map) and separate lines of research have focused on deep learning (e.g., Transformers, MLP-based approaches), no previous study has explicitly integrated these two strands under a single, end-to-end framework.

**Theoretical Claims:**

There are no proofs to verify beyond the statements of Takens’ theorem, which the authors cite rather than prove themselves.

---

> ### Author Rebuttal · Authors · 2025-04-01
>
> Thanks for your detailed feedback. We’re pleased that you recognize the novelty of combining Empirical Dynamical Modeling with deep learning and the strong performance of DeepEDM, especially in noisy settings. We also appreciate your acknowledgment of our connection to the Attention mechanism and the flexibility of our approach. Below, we address your questions.
>
> ### 1. Validation of Latent Space Representation
>
> Great question! The latent space in DeepEDM functions as a learned kernel. We hypothesize that with noise, the learned kernel retrieves nearest neighbors more faithfully in the underlying state space compared to the time-delay embedding, which, as dictated by Takens' theorem, relies on nearly noise-free data for accurate reconstruction. This is indirectly supported by our results in Table 6, where DeepEDM outperforms EDM in time series forecasting.
>
> To provide a more direct validation, we design and conduct a new experiment using simulated time series generated from the Lorenz system, where true neighbors in the state space are known. We evaluate the ability to retrieve these neighbors using different distance metrics. Specifically, for a given state at time $t$, we first determine $k$ true neighbors in the original state space. For the same data point at time $t$, we then retrieve the top $m$ nearest neighbors in (i) the time-delay embedded space using Euclidean distance, and (ii) DeepEDM's latent space using the learned kernel. For each query point, we evaluate the recall of the retrieved neighbors, quantifying the fraction of true neighbors successfully retrieved. For a comprehensive evaluation across different settings, we conduct this experiment with varying values of $m \in \{7, 14, 28\}$, fixing $k = 7$ (the minimum number of neighbors necessary for recovering the dynamics). We also consider two noise conditions: no noise ($\sigma = 0$) and w. noise ($\sigma = 2.5$). A high recall score indicates a distance metric can successfully identify true neighbors in the state space necessary to reconstruct the dynamics. The results of this experiment can be found here:
>
> https://anonymous.4open.science/r/icml_rebuttal-B440/nearest_neighbor_retrieval_exp_results.md
>
> As expected, under noise-free conditions, both methods perform similarly. However, when noise is added, the vanilla time-delayed embeddings exhibit a sharp decline in recall. In contrast, our learned kernel in DeepEDM degrades more gracefully, maintaining more accurate retrieval of neighbors. This suggests that the state space reconstructed using the learned kernel more accurately reflects the underlying state space, as it preserves the local neighborhood structure.
>
> ### 2. Generalization as a Unified Parametric Model
>
> Please refer to our response #1 to reviewer Ywcy.
>
> ### 3. Selection of baselines
>
> We respectfully disagree with the comment, 'baselines (for synthetic dataset experiments) are predominantly EDM-based'. The following baselines were considered (for this experiment) in the paper:
>
> - **Simplex**: A classical EDM approach, as the starting point of DeepEDM.
> - **Koopa**: A recent deep model based on Koopman operator, as a representative method that combines deep learning and dynamical system modeling (Koopman).
> - **iTransformer**: A recent deep learning-based model, as a representative method that is pure learning based using a Transformer.
>
> This selection ensures a well-rounded comparison against strong and methodologically diverse baselines. We note that the full set of results using these baselines is reported in Table 6 of the Appendix.
>
> ### 4. Missing References on Chaotic Time Series Forecasting
>
> Your suggestion to incorporate prior work on chaotic time series forecasting is well taken. While our focus has been on EDM and deep learning-based forecasting, we recognize the relevance of research in chaotic time series forecasting. We drafted the following paragraph to add to our related work section. We welcome your further feedback.
>
> *A related research direction focuses on forecasting chaotic time series via state space reconstruction, mirroring the underlying principles of EDM. Pioneering work by Farmer & Sidorowich (Phys. Rev. Lett. 1987) introduced local approximation techniques within reconstructed state spaces using delay embeddings, facilitating short-term predictions. Subsequent studies explored the application of feedforward neural networks for learning direct mappings from reconstructed phase states to future states (Karunasinghe & Liong Journal of Hydrology 2006). Recurrent neural networks, particularly Echo State Networks (ESNs), have also shown promise, with adaptations like robust ESNs (Li, Hand & Wang. IEEE TNNLS 2012) addressing the inherent sensitivity of chaotic signals to noise and outliers. However, a significant gap remains: the development of a fully differentiable, end-to-end trainable neural network architecture that seamlessly integrates dynamical systems theory with deep learning methodologies.*

---

> > ### Comment · Reviewer_XFih · 2025-04-09
> >
> > Thanks for the detailed responses and the additional experiment on latent space validation. I found that particularly helpful. The new paragraph on prior chaotic time series work is also a welcome addition.

---

> > > ### Author Response · Authors · 2025-04-09
> > >
> > > Dear Reviewer,
> > >
> > > Thank you for your thoughtful follow-up and for raising the score. We are pleased that you found the latent space validation experiment helpful. We are also grateful to you for pointing out the missing related work, and we will be sure to include the paragraph in the final paper.
> > >
> > > Since you recognized the novelty and the thorough comparison on standard benchmarks of our work, further noting it [appears to] be the first to systematically merge Empirical Dynamical Modeling with deep neural architectures in an end to end framework, we would greatly appreciate any further reconsideration of the score if you feel the paper merits it.
> > >
> > > Regards,
> > >
> > > Authors

---

### Official Review · Reviewer_qg1c · 2025-03-13

**Overall Recommendation:** 3

**Summary:**

This paper introduces DeepEDM, a framework that integrates nonlinear dynamical systems modeling with deep neural networks for time series forecasting. Built on empirical dynamic modeling (EDM) and Takens' theorem, DeepEDM employs time-delayed embeddings in a latent space and uses kernel regression to approximate underlying dynamics. This approach aims to enhance forecasting accuracy by explicitly modeling complex time series dynamics.

**Claims And Evidence:**

Overall, the claims made in the paper are clear.

**Essential References Not Discussed:**

The paper appropriately discusses related work, and I did not identify any missing essential references.

**Experimental Designs Or Analyses:**

I have reviewed the experimental design and analyses, and they appear to be sound.

**Methods And Evaluation Criteria:**

Yes, the proposed methods and evaluation criteria are appropriate for the problem of time series forecasting.

**Other Comments Or Suggestions:**

N/A

**Other Strengths And Weaknesses:**

Strengths
1. The paper is well-organized, making it easy to follow the motivation, methodology, and results.
2. The integration of empirical dynamic modeling (EDM) with deep learning for time series forecasting is novel and addresses the challenge of capturing nonlinear dynamics.
3. The proposed approach is supported by extensive experiments and analyses, demonstrating its effectiveness across multiple datasets.

Weaknesses
1. The paper claims that DeepEDM mitigates EDM's sensitivity to noise and is potentially noise-free. However, the underlying mechanism for this claim is not explicitly clear. Further clarification on how DeepEDM addresses this limitation would strengthen the argument.
2. The proposed method relies on the time delay hyperparameter, which is crucial in EDM-based approaches. How is the time delay selected in practice? Additional ablation studies on the effect of time delay and embedding dimension would be beneficial.
3. The method underperforms on certain datasets, particularly those with a high number of variables (e.g., Electricity, Traffic). Providing an explanation for this performance drop would improve transparency. Additionally, results using a more standard MSE-based objective would help verify that the reported performance gains are due to model improvements rather than loss function optimization.

**Questions For Authors:**

Please refer to the Weaknesses.

**Relation To Broader Scientific Literature:**

The paper contributes to the broader scientific literature by integrating nonlinear dynamical systems modeling with deep learning for time series forecasting.

**Theoretical Claims:**

The paper presents theoretical claims related to nonlinear dynamical systems and their integration with deep learning. While I have reviewed the theoretical justifications, I am not fully confident in the correctness of all derivations.

---

> ### Author Rebuttal · Authors · 2025-04-01
>
> Thank you for your valuable feedback and recognition of our paper’s clarity, structure, and experimental design. We appreciate the acknowledgment of our contribution to nonlinear dynamical systems through EDM-integrated deep learning. Below, we address your suggestions and will incorporate these responses into the final version.
>
> ### 1. Noise Robustness of DeepEDM
>
> We hypothesize that DeepEDM’s noise robustness stems from its learned kernel, which enables adaptive distance assessments for nearest-neighbor retrieval, rather than using the fixed Euclidean distance in traditional time-delay embeddings. Fixed metrics are more sensitive to perturbations, while the learned kernel adapts to the data, thus improving neighbor identification under noise.
>
> To validate this, we conducted additional experiments (please refer to our response #1 to Reviewer XFih for more details) on nearest-neighbor retrieval. In short, we compare (i) the time-delay embedded space using Euclidean distance, and (ii) DeepEDM's latent space using the learned kernel for retrieving true neighbors in the state space using synthetic data with known dynamics. Our results show that the learned kernel in DeepEDM achieves higher recall under noise, confirming that the learned kernel indeed enhances robustness against noise. This robustness is further supported by the superior forecasting performance of DeepEDM compared to EDM w. Simplex, under noisy conditions (see Table 6).
>
> ###  2. Sensitivity to Time Delay and Embedding Dimension
>
> We conducted additional experiments to study the effects of time delay and embedding dimensions. Specifically, DeepEDM has two parameters that control time delay and embedding dimensions. First, $m$ defines the number of time-delayed steps, and $\tau$ controls the interval between these steps. For example, $m=3$ and $\tau=1$ results in a 3-dimensional embedding $[y_t, y_{t-1}, y_{t-2}]$, while $m=3$ and $\tau=2$ would yield an embedding $[y_t, y_{t-2}, y_{t-4}]$. In our work, $m$ is determined empirically for each dataset, while $\tau$ is set to 1. Following your suggestion, we conduct ablation experiments to analyze the impact of these hyperparameters. For the experiments on $m$, we fix $\tau = 1$ and vary $m$ over $[3, 5, 7, 11, 15]$. Conversely, for the experiments on $\tau$, we set $m = 5$ and explore $\tau$ in $[1, 2, 3]$. The results are linked below.
>
> **Effects of time-delayed steps $m$ (i.e., embedding dimension)**: Our findings indicate that the impact of $m$ is dataset-dependent. In some cases, performance remains relatively stable across different values of $m$, suggesting that the intrinsic state-space dimensionality might either be very low or very high. If the state-space dimension is small, larger values of $m$ will capture the underlying dynamics and thus perform similarly well. On the other hand, if the state-space dimension is large, different small values of $m$ will not be able to model the dynamics, yielding results at a similar performance level. This behavior aligns with the intuition that effective embedding reconstruction depends on the underlying system complexity.
>
> https://anonymous.4open.science/r/icml_rebuttal-B440/m_delay_results.md
>
> **Effects of delay interval $\tau$:** Our results show that in most cases, $\tau = 1$ yields the best performance, suggesting that a small stride is sufficient. This is also the case considered in Takens’ theorem. However, it remains possible that an optimal balance between $\tau$ and $m$ could further improve results, which we leave for future investigation.
>
> https://anonymous.4open.science/r/icml_rebuttal-B440/tau_results.md
>
> ###  3. Performance degradation on High-Dimensional Datasets
>
> Please refer to our response to Q5 from Reviewer TNFN.
>
> ###  4. Loss Function Impact
>
> Our ablation study (Table 4) shows that our DeepEDM significantly enhances forecasting performance (MLP vs. MLP+EDM, without any loss function optimization) and the loss optimization further reduces the errors (MLP+EDM w/o TDT loss vs. Full Model w. TDT loss). To further delineate the contributions of DeepEDM and loss optimization, we conducted an additional experiment. Specifically, we compare the DeepEDM trained with standard MSE loss against DeepEDM trained with the optimization objective incorporated in our paper. The results of this experiment are available at the table linked below. Notably, the differences remain marginal—while in some cases, the MSE loss further improves DeepEDM’s performance, in others, it results in slight declines.
>
> https://anonymous.4open.science/r/icml_rebuttal-B440/loss_exp_results.md

---

> > ### Comment · Reviewer_qg1c · 2025-04-08
> >
> > Thank you to the authors for their detailed and thoughtful responses. I appreciate the clarifications and additional results provided, which help address several of the concerns I raised in my initial review.
> >
> > While the rebuttal has clarified some points and improved my understanding of the work, it does not substantially shift my overall assessment. Therefore, I will maintain my original score.

---

> > > ### Author Response · Authors · 2025-04-09
> > >
> > > Dear Reviewer,
> > >
> > > Thank you for your thoughtful response and for taking the time to review our rebuttal. We are pleased to hear that our rebuttal has addressed several of your concerns, and we appreciate the opportunity to further clarify our work. We deeply value your assessment.
> > >
> > > As you kindly noted, our rebuttal has addressed several of your concerns. We also believe that the additional experiments and clarifications (specific to your review, as well as others in the rebuttal) address weaknesses #1, #2, and #3 that you highlighted, while further strengthening the contributions of our work. Given these updates, as well as your initial recognition of the novelty of our work, the well-organized nature of our paper, and our extensive empirical analyses, we would be grateful if you could please consider raising your score. If there are any remaining concerns or points that still require attention, we welcome your insights and would be happy to incorporate any feedback or suggestions you may have.
> > >
> > > Thank you once again for your valuable feedback and continued engagement with our work.
> > >
> > > Best regards,
> > >
> > > Authors

---

### Official Review · Reviewer_TNFN · 2025-03-14

**Overall Recommendation:** 2

**Summary:**

The paper resorts to first principles, to examine the usefulness of using embedology (as in Takens' embedding theorem) in conjunction with neural networks. This is important and has been missing in the literature.

**Claims And Evidence:**

DeepEDM claims three key advantages.
(1) By learning a latent space of the time-delayed embeddings, it mitigates
the sensitivity of EDM to input noise;
(2) Unlike EDM,which requires a separate model for each time series, it
learns a single parametric model that generalizes to new
time series; and
 (3) It offers flexibility and scalability, providing theoretic insights for Transformer-based
time series forecasting models

**Essential References Not Discussed:**

N/A

**Experimental Designs Or Analyses:**

The approach is evaluated on both synthetic and real world time series. The synthetic series involve the Lorenz and Rossler chaotic signals (which exhibit clear attractors).

**Methods And Evaluation Criteria:**

The integration of dynamical systems theory with time series forecasting is important and relatively under-explored.

**Other Comments Or Suggestions:**

None

**Other Strengths And Weaknesses:**

Strentghs:
- the paper introduces a theoretical framework, but the exepriments do not fully justify the approach
- The empirical results on real world data are good but not consistently outperforming SOTA.
- the proposed method is not very sensitive to the lookback length (time segment size), which is unexpected


Weaknesses:
- are the curves in Fig 2plotted in the absolute terms or in a dB scale?
- There is little consistency in the empirical analysis.
- Is the proposed method better suited for some kinds of time series than for other kinds?
- the simulations vs lookback length in Fig 4 in the supplement are important and better suited to be part of the main paper.
- since the paper deals with multivatiate time series, I would have expected deep analysis of the performance wrt the number of variates in a multivariate time series
- the performance under noise is not convincing

**Questions For Authors:**

See Weaknesses

**Relation To Broader Scientific Literature:**

The paper extends and combines two classical approaches, within the framework of transformers:
-  Empirical Dynamical Modeling (EDM) (Sugihara & May, 1990),
-  Takens’ embedding theorem (Takens, 1981; Sauer et al., 1991),

**Theoretical Claims:**

The paper employs an efficient attention-based kernel regression, as a general framework to consider Transformer-based time series models. It maps time-delay embeddings, derived from Takens’ theorem, into a learnable, higher-dimensional latent space that is potentially noise-free, facilitating a more precise reconstruction of the system’s underlying dynamics. DeepEDM can be viewed as employing a modified cross-attention framework. Similar to cross attention, the focal vectors act as queries (Q), the historical data points (Y) serve as keys (K), while the future states (y + ∆t) act as the values (V). Further, the relationship between queries and keys is quantified using the dot product as the similarity metric modulated by θ. However, DeepEDM diverges from traditional attention mechanisms by enforcing a structural differentiation where keys and values are explicitly distinct.

I am famialiar with the Takens type of embedding and in my opinion the approach in this paper is correct.

---

> ### Author Rebuttal · Authors · 2025-04-01
>
> Thank you for your detailed feedback and recognition of our theoretical contributions. We appreciate your acknowledgment that embedology, despite its foundational role, has been largely overlooked and that our work helps bridge this gap by integrating Takens' embedding theorem with neural networks. Your recognition of our approach as a step toward extending EDM and providing theoretical insights into Transformer-based forecasting is also greatly valued. Below, we address your suggestions and will integrate these responses into the final version.
>
> ###  1. Clarification on Figure 2B
>
> The axis in Figure 2B is presented in absolute terms, which we will clarify in the final version. We chose to not normalize the data (e.g. mean subtraction, standardization) to preserve the raw performance of the model. While this increases error ranges—especially in chaotic Lorenz settings where noise sensitivity is high—it provides a more transparent evaluation. Detailed results are in Appendix A.7.
>
> ###  2. Consistency of Empirical Analysis
>
> Our empirical analysis consists of experiments and results using (a) synthetic time series generated from known, challenging dynamic systems (Lorenz and Rossler); and (2) real-world time series from multiple application domains. For synthetic data, we demonstrated that DeepEDM is more robust to noise when compared to the vanilla EDM, and is more accurate when compared to other representative learning-based methods (Koopa and iTransformer). For real-world data, we showed that DeepEDM consistently outperforms prior methods across different settings (e.g., fixed vs. varying lookback length). This consistency was evidenced by the overall ranking of our method (e.g., in Table 1, DeepEDM ranks the top 1 for 36 out of 72 metrics while the next best model achieves 11) as well as the results of multiple runs (Table 5 in the Appendix). Additionally, we presented extensive ablation studies of our design and lookback length. We believe our current empirical analysis is sufficient to support the main claim of the paper. Yet we welcome specific suggestions to improve our empirical analysis.
>
> ### 3. Sensitivity to lookback length
>
> Theoretically, Takens’ theorem puts constraints over the number of time-delayed steps (>2d for d-dimensional state space), yet does not speak to the number of samples needed (i.e., lookback length). Empirically,  we observe that the model exhibits minimal sensitivity to the lookback length beyond a certain threshold. Once the model has sufficient historical data, additional time steps—particularly those with similar dynamics—contribute little to improving forecast performance. On the other hand, too short a lookback window can degrade performance, as there may be insufficient information to generate an accurate forecast. This observation is shown in our paper: the existence of an optimal lookback length, or "sweet spot," as illustrated in Figure 4.
>
> ###  4. Moving Figure 4 to the main paper
>
> Given the additional one-page allowance in the final version, we plan to incorporate this analysis into the main paper.
>
> ###  5. Benefit of DeepEDM for Some Kinds of Time Series than for Others
>
> Great question! We will include a discussion in the main paper. Our main findings from the empirical analysis are twofold. First, there is a sweet spot in the lookback length, which is dataset-dependent (discussed in our response #3). Second, DeepEDM is best suited to time series with a low to moderate number of input variates, as seen in datasets like the ETT set, Weather, ILI, and our simulated data. For datasets such as Traffic and ECL, which contain a larger number of variates, we observe a slight decline in performance. One possible reason is that DeepEDM considers a channel-wise prediction using a shared model across all channels, and thus assumes independence between input variates. With a growing number of variates, there is likely an increasing strength of correlation among these variates, which our DeepEDM can not effectively model. We plan to further explore this correlation among input variates in future work.
>
> ###  6. Performance Under Noise
>
> We highlight that noisy conditions present a significant challenge for forecasting models, leading to performance degradation in strong baselines such as iTransformer, Koopa, and the vanilla Simplex. However, DeepEDM demonstrates greater robustness, exhibiting less performance degradation compared to these baselines. As discussed in Sec 5.1 and further shown in Table 6, DeepEDM achieves the best performance among strong baselines in the presence of noise. These results support our claim on noise robustness, as also noted by reviewer XFih that “[DeepEDM] remains robust even in the presence of noisy observations”.

---

### Official Review · Reviewer_Ywcy · 2025-03-20

**Overall Recommendation:** 3

**Summary:**

This paper introduces a new framework, DeepEDm, for time series forecasting, inspired by studies in Empirical Dynamic Modeling (EDM). The approach is fundamentally based on Taken's theorem, which states that a dynamic system can be reconstructed using a delay embedding of the series in phase space.

In practice, the method starts by constructing a time-delay representation of the series, from which an embedding is learned in a latent space. Forecasting is then performed using a regression kernel, which assigns weights to the closest representations from the training set in the latent space. Finally, a classic MLP decoder generates the forecast.

Experiments on both synthetic and real-world datasets demonstrate the performance of the proposed model.

**Claims And Evidence:**

The main claim is that the proposed method achieves strong performance compared to the state of the art and is theoretically grounded. The experimental campaign indeed demonstrates the effectiveness of the proposed approach.
A secondary claim is mentioned in the introduction: "it learns a single parametric model that generalizes to new time series." However, I am not entirely sure what the authors mean by this statement. My understanding might be incorrect, but I expected to see experiments on adaptation or transfer to previously unseen time series in the datasets. However, the paper does not include any experiments addressing this issue.

**Essential References Not Discussed:**

NA

**Experimental Designs Or Analyses:**

The paper follows recent publications in terms of benchmark selection and evaluation protocols. However, the benchmarks commonly used in the field are known to be simple and highly stable.

The authors chose to present results in the main body of the paper based on a very long lookback window (twice the forecasting horizon), while results for other lookback windows are only provided in the appendix. However, these results are not detailed per lookback window; instead, only the best result is displayed (Table 3). I believe these results should be presented in greater detail, with a breakdown by lookback window size.

Additionally, Table 4 (ablation study) reports averaged results, which somewhat obscures key insights. Specifically, it shows that a simple linear model achieves results very close to both the baselines and the proposed model, on average, across all forecasting horizons. To better understand the challenge posed by the task and the added value of more complex methods over a simple linear model, these results should be detailed more thoroughly.

It is unfortunate that there are no experiments on transfer learning or domain adaptation. The proposed method seems well-suited for such tasks, given the learned latent space, and this could have been a key strength of the approach.

**Methods And Evaluation Criteria:**

The selected benchmarks are standard in the field, although they are known for their simplicity and stability. The two evaluation metrics used—MAE and MSE—are widely used in forecasting. The chosen baselines are also comprehensive, ensuring broad coverage of existing architectures

**Other Comments Or Suggestions:**

As mentioned above, it would be beneficial to provide detailed experiments for the different lookback windows and the ablation study. Additionally, it would be valuable to include experiments on more competitive tasks.

**Other Strengths And Weaknesses:**

NA

**Questions For Authors:**

NA

**Relation To Broader Scientific Literature:**

The paper is in line with recent works in time series forecasting, but it shares the same limitations, namely the use of simple benchmarks that may no longer be suitable.
The paper also draws a connection between the framework used and self-attention, which is indeed very similar once the latent projection method is defined.

**Theoretical Claims:**

The formalism is derived from Empirical Dynamic Modeling (EDM). It is clearly presented and easy to understand, with a summary of the mathematical tools provided in the appendix. The paper does not include any additional theoretical elements that require validation.

---

> ### Author Rebuttal · Authors · 2025-04-01
>
> Thank you for your thoughtful and detailed review. We now address your suggestions. Our response will be integrated into the final version.
>
> ### 1. Clarification on Generalization Statement
>
> This was meant to contrast DeepEDM and EDM w. simplex: EDM trains a separate model for each time series, whereas DeepEDM learns a single shared model across all series (within same dataset). The statement is partially supported by our main results (Table 1), where models were evaluated on unseen time series. In this widely-used evaluation protocol, the training series and test ones may be extracted from the same long sequence. Due to this design, we realize that part of this statement (“generalization to new time series”) is ambiguous and will revise it in the final version.
>
> To further clarify, we conducted an additional experiment on ETT, Exchange, and Weather datasets to evaluate DeepEDM's ability to generalize across “different” time series (within the same dataset). Specifically, for each dataset, we trained the model on a subset of sequences (seqs.) and tested it on a different subset.
> - For the ETT datasets, which consist of 7 seqs., we trained on the seq. 0-2 (using only timesteps from the standard training split) and tested on seq. 4-6 (using timesteps from the standard test split). We use this 3:3 split as several baseline models are unable to handle different channel dimensions between training and testing.
> - Similarly, for the Exchange dataset (8 seqs.), we trained on the first 4 sequences and tested on the latter 4.
> - For the Weather dataset (21 seqs.), we trained on sequences 0-9 and tested on sequences 10-19.
>
> We compared DeepEDM against strong baselines including Koopa, iTransformer, and PatchTST. The prediction horizon (H) varied from [48, 96, 144, 192] and lookback window was set to $2*H$. The summarized results are presented below along with the link to full results. Notably, DeepEDM ranks 1st in MAE and MSE winning in 39 out of 48 cases. The results thus provide more direct support to the statement of “a single model that generalizes to new time series."
>
> | Dataset   | DeepEDM (MSE/MAE)     | Koopa (MSE/MAE)   | PatchTST (MSE/MAE)    | iTransformer (MSE/MAE)   |
> |:----------|:----------------------|:------------------|:----------------------|:-------------------------|
> | ETTh1     | (**0.230**/**0.310**) | (0.248/0.330)     | (0.253/0.330)         | (0.259/0.330)            |
> | ETTh2     | (**0.151**/**0.230**) | (0.170/0.260)     | (0.183/0.270)         | (0.170/0.250)            |
> | ETTm1     | (**0.214**/**0.280**) | (0.242/0.310)     | (0.216/0.290)         | (0.245/0.300)            |
> | ETTm2     | (**0.075**/**0.160**) | (0.086/0.180)     | (0.088/0.180)         | (0.096/0.190)            |
> | Exchange  | (0.106/**0.200**)     | (0.146/0.240)     | (**0.105**/0.210)     | (0.131/0.230)            |
> | Weather   | (**0.293**/**0.250**) | (0.325/0.280)     | (**0.293**/**0.250**) | (0.423/0.320)            |
>
> https://anonymous.4open.science/r/icml_rebuttal-B440/generalization_exp_results.md
>
> ###   2. Lookback Window Reporting
>
> Since historical data is typically accessible, any reasonable lookback window can be assumed and the lookback length is often treated as a hyperparameter [Bergmeir, 2023]. Therefore, we tuned lookback length separately for each model and forecasting horizon, and only reported best results without a detailed breakdown. This protocol was also considered in prior works [TimeMixer, PatchTST, DLinear]. The effects of lookback length in forecasting performance was indeed discussed in Appendix A.5.2 (Figure 4). If needed, we can add a detailed breakdown to the Appendix.
>
> ###  3. Ablation Study Reporting and Results of Linear Models
>
> Our ablation study is designed to concisely highlight the method design choices. These detailed experiments span 9 datasets, 4 lookback settings, and 4 configurations with multiple runs. To maintain clarity and conciseness, we presented aggregated results describing key trends. Per reviewer’s request, we have included the full table below, which we will add to the Appendix.
>
> https://anonymous.4open.science/r/icml_rebuttal-B440/ablation_full_results.md
>
> **Linear models**:
> As prior work  [DLinear, RLinear] has noted, linear models are effective for forecasting, particularly for simpler datasets. However, they struggle with more complex datasets. As shown in the above table, DeepEDM significantly outperforms the linear model on benchmarks like ILI, Weather, ECL, and Traffic by a large margin, e.g., on ECL the relative improvement in MSE from the linear model to our method is ~10%.
>
> ###  3. Transfer learning / domain adaptation
>
> This is an exciting future direction! Our work focuses on single-domain forecasting but shows transfer within-domain (see Q1). Cross-domain transfer learning requires further study and we leave it as future work.

---

### Decision · Program_Chairs · 2025-05-01

**Decision:**

Accept (poster)

**Comment:**

The paper introduces DeepEDM, a framework that integrates nonlinear dynamical systems modeling with deep neural networks for time series forecasting. The authors have addressed the concerns from the reviewers. Most of the reviewers support the paper. Please incorporate all the discussions into the final version of the paper.